



# Aerosol-cloud interactions in mixed-phase convective clouds. Part 1: Aerosol perturbations.

Annette K. Miltenberger[1], Paul R. Field[1,2], Adrian A. Hill[2], Phil Rosenberg[1], Ben J. Shipway[2], Jonathan M. Wilkinson[2], Robert Scovell[2], and Alan M. Blyth[3]

[1]Institute of Climate and Atmospheric Science, School of Earth and Environment, University of Leeds, United Kingdom
[2]Met Office, Exeter, United Kingdom
[3]National Centre for Atmospheric Science, School of Earth and Environment, University of Leeds, United Kingdom

*Correspondence to:* Annette K. Miltenberger (a.miltenberger@leeds.ac.uk)

**Abstract.** Changes induced by perturbed aerosol conditions in moderately deep (cloud top at about $5\,\mathrm{km}$) mixed-phase convective clouds developing along sea-breeze convergence lines are investigated with high-resolution numerical model simulations (grid spacing of $250\,\mathrm{m}$). The simulations utilise the newly developed Cloud-AeroSol Interacting Microphysics module (CASIM) for the Unified Model, which allows for the representation of the two-way interaction between cloud and aerosol

fields. Simulations are evaluated against observations collected during the COPE field campaign over the southwestern peninsula of the UK in 2013. The simulations compare favourably with observed thermodynamic profiles, cloud-base cloud droplet number concentrations (CDNC), cloud depth, and radar reflectivity statistics. Including the modification of aerosol fields by cloud microphysical processes in the simulations improves the match to observed cloud-base CDNC, increases the CDNC variability and leads to a larger decrease of CDNC with height above cloud base. However, it also reduces the average cloud

size and cloud top height, which is less compatible with observations.

Before clouds become organised along the sea-breeze convergence lines, precipitation is suppressed by increasing aerosol due to less efficient precipitation production by warm-phase microphysics. The precipitation suppression is less evident if aerosol processing is taken into account. After the sea breeze convergence zone is established, accumulated precipitation from the on average deeper and wider clouds increases with aerosol concentrations as long as cloud top heights are not limited by an

upper level stable layer. The precipitation enhancement is controlled by changes in condensate production and precipitation efficiency. Enhanced condensate production in high aerosol scenarios is related to higher vertical velocities in the convective cores, i.e., convective invigoration, and stronger latent heating below the $0\,^{\circ}\mathrm{C}$ level, while changes in latent heating in the mixed-phase region are negligible. Perturbed aerosol concentrations alter the cloud field structure with fewer larger cells developing in high aerosol environments, but inducing only small changes in cloud fraction. It is hypothesised that the stronger

latent heating from convection is related to the changes in the cloud field structure reducing the mixing of environmental air into the convective core. For very high aerosol concentrations, the translation of convective invigoration into deeper clouds and enhanced precipitation is limited by thermodynamic constraints.

The aerosol-induced changes in shallow warm-phase clouds prior to the development of a strong convergence zone is consistent with ideas based on parcel models. However, the precipitation response of the deeper mixed-phase clouds along well-

established convergence lines suggest that when clouds begin to interact with the pre-existing thermodynamic environment



and modifications to the cloud field structure occur, i.e., processes other than microphysics effect the cloud evolution, and the precipitation behaviour can be opposite to predictions from parcel models.

*Copyright statement.* TEXT

# 1 Introduction

The aerosol-induced changes to the climate system, in particular the radiation budget, are thought to be important to understand changes between present-day and pre-industrial radiative forcing (Stocker et al., 2013). A large and poorly constrained aspect is the impact of aerosols on clouds and precipitation formation (Stocker et al., 2013). Numerous studies have tried to isolate the aerosol effect on clouds and precipitation using observational data or investigated the aerosol effect in numerical models of varying complexity. Recent reviews by Khain (2009), Tao et al. (2012), Altaratz et al. (2014) and Rosenfeld et al. (2014)

provide a good overview.

The concept of aerosol-induced changes to clouds is based on the well-established link between the number of aerosols available and the number of cloud droplets that form under a specific supersaturation. This initial change in cloud droplet number should have knock-on effects on radiative and cloud microphysical processes, which are directly dependent on the number and size of involved hydrometeors. These impacts can have further ramifications by altering the precipitation formation in clouds,

their geometry and lifetime, anvil properties, thermodynamic properties of the environment and the spatial pattern of energy and moisture transport. In the real atmosphere, a multitude of other processes such as interactions between different clouds, aerosol physics, aerosol-radiation interactions, larger-scale dynamics and surface fluxes can further complicate the picture. While the first link in this chain, the relation between aerosol concentration and cloud droplet number at cloud base, is un-controversial and can be confirmed with observational data (e.g., Andreae, 2009), the subsequent impacts on the temporal and

spatial evolution of the cloud field are more controversial and difficult to observe. Perhaps, unsurprisingly given the complexity, highly non-linear nature and our partial quantitative understanding of many of the processes involved, different studies do not necessarily agree on the amplitude or sign of the impacts. Some attempts have been made to assess more systematically the impact of specific model parameters (e.g., Johnson et al., 2015) or to stratify responses according to crucial meteorological parameters (e.g., Khain and Lynn, 2009; Altaratz et al., 2014). Khain and Lynn (2009) use the concept of expressing changes in

precipitation as the result of modified condensation production by lifting ($\Delta$G) and modified evaporation losses of condensate ($\Delta$L). With this approach they are able to classify aerosol-induced precipitation changes documented in various observational and modelling studies. According to their analysis the balance between $\Delta$G and $\Delta$L is dependent on the cloud regime and environmental conditions. For example, $\Delta$L dominates in stratocumulus and $\Delta$G in deep tropical clouds, while deep convective clouds transition from $\Delta$L to $\Delta$G dominated with increasing environmental relative humidity.

The increased short-wave reflectance (e.g., Twomey, 1977) and decreased efficiency of collision-coalescence as a consequence of higher cloud droplet number concentrations are thought to dominate aerosol-cloud interactions in shallow warm-phase





clouds. The reduced collision-coalescence delays or suppresses precipitation formation and extends the cloud lifetime (e.g., Albrecht, 1989; Lohmann and Feichter, 2005). For deep convective clouds with partially or completely glaciated cloud tops, it has been hypothesised that precipitation can be enhanced or invigorated through feedbacks on the cloud dynamics (Khain et al., 2005; Koren et al., 2005; Rosenfeld et al., 2008). The main idea is that a slower growth of cloud droplets into precipitation-sized

particles in the warm-phase section of the cloud enhances the transport of cloud condensate into the mixed-phase region. The additional latent heat release from the subsequent freezing of this additional condensate leads to a higher cloud top, a larger total condensate content, a larger anvil and a longer cloud lifetime. These changes, together with accompanying modifications of the precipitation production pathways (e.g., Wang, 2005; Fan et al., 2007; Li et al., 2009; Cui et al., 2011), are hypothesised to lead to a precipitation enhancement under high aerosol conditions. This conceptual idea has been developed using simulations

of individual clouds under idealised conditions. Several studies have highlighted that this may not apply to less idealised or very polluted conditions due to a number of factors including an increased importance of evaporation and stronger downdrafts (Lebo and Seinfeld, 2011). A further important aspect that may impede such a mechanism is a weakening of the updraft core by the increased water loading (e.g., Seifert and Beheng, 2006; Lebo and Seinfeld, 2011). Also, Johnson et al. (2015) demonstrated that the precipitation signal is dependent on the values of uncertain parameters within the cloud microphysics parameterisation

(parameter uncertainty). Changes in cloud properties and precipitation are also subject to systematic differences between and biases in different model formulations including the formulation of parameterisations, dynamics and spatial resolution (e.g., Lebo and Seinfeld, 2011; Lebo et al., 2012; Fan et al., 2012; Morrison, 2012; Hill et al., 2015; White et al., 2016).

Precipitation enhancement and/or associated changes in the cloud structure are not always predicted consistently for different cases even within the same modelling framework. This illustrates that different environmental conditions and interactions be-

tween different clouds (direct or indirect via modification of the environment) can influence, impede or allow for precipitation enhancement (e.g., van den Heever et al., 2006; Tao et al., 2007; Fan et al., 2009; Khain and Lynn, 2009; Fan et al., 2012; Lebo and Morrison, 2014). Aerosol-induced changes of cloud properties can impact the thermodynamic and aerosol environment (e.g., Lee and Feingold, 2010; Cui et al., 2011; Morrison and Grabowski, 2011) and storm-scale (e.g., Lebo and Morrison, 2014) or even large-scale dynamics (e.g., Lee, 2012), which are then often found to modify the aerosol impact on the entire

cloud system. The precipitation feedback is typically less clear and more sensitive to the particular microphysical scheme used than changes in cloud radiative properties. For example, in a study of convective precipitation during three summer seasons over Germany Seifert et al. (2012) found that the aerosol induced modification of surface precipitation by aerosols were negligible, but the modifications to cloud radiative properties were not.

Given the hypothesised importance of aerosol-cloud interactions for radiative forcing estimates and quantitative precipitation

forecasts and the uncertainties and deficiencies of numerical models, it is important to test any model derived hypothesis with observational data. A number of observational studies have tried to identify an aerosol-signal in the properties of deep convective systems, including systematic changes in cloud top height, cloud fraction or precipitation (e.g., Devasthale et al., 2005; Koren et al., 2010; Gryspeerdt et al., 2014). These studies are based on satellite data, which provide a relatively large temporal and spatial sample. Studies based on satellite data necessarily rely on correlations between bulk parameters such as aerosol op-

tical depth and cloud top height or fraction, which raises the question of causality, coincidence and co-variability (e.g., Stevens



and Feingold, 2009). The need to better understand and incorporate the existence of co-variability between aerosol and mete-orological fields in analysis methods has recently been highlighted by Feingold et al. (2016). In this context, it is important to consider how similar, in a meteorological sense, different instances must be for a meaningful analysis and whether the analysis of a sufficiently large sample (how large?) provides a robust cloud-aerosol signal. One approach to address the questions of the

implications of co-variability from modelling standpoint is to use an ensemble forecasting system.

In this study we use a convection-permitting numerical weather prediction model (the Unified Model) with a multi-moment bulk microphysics scheme to investigate the aerosol-cloud interactions in an observed case of mixed-phase convective clouds forming along a sea-breeze convergence zone. Sea-breeze convergence zones provide a predictable location for convective initiation, which aids the comparison to observations and also provided a good basis for planning observational campaigns.

Convective clouds and precipitation are associated with sea-breeze systems at many coastal regions on the globe, including e.g., the southwest peninsula of the UK (e.g., Golding et al., 2005), the Salento Peninsula in Italy (e.g., Comin et al., 2015), the Hainan Island in China (e.g., Liang and Wang, 2016), coastal Cameroon (e.g., Grant and van den Heever, 2014) and many others (e.g., Miller et al., 2003).

The case being considered is from the COPE campaign, which took place in July and August 2013 over the southwestern

peninsula of UK (Leon et al., 2016; Blyth et al., 2015). The selected case (3$^{rd}$ August 2013) has been previously analysed from an observational viewpoint with a focus on cloud glaciation (Taylor et al., 2016b) and aerosol concentrations, composition, and sources (Taylor et al., 2016a). Isolated shallow cumulus clouds were scattered across most of the southwestern UK in the early morning. Two sea-breeze fronts developed along the northern and southern coastlines of the peninsula and propagated inland. After about 11 UTC clouds organised along convergence lines, which were located roughly along the major axis of

the peninsula. The organisation proceeds with the development of larger and on average deeper clouds and cloud clusters. Clouds were arranged in two lines with a latitudinal offset of about 20-30 km. New, isolated cells generally developed close to the south-western tip of the peninsula and subsequently developed or merged into larger clouds clusters as they move north-eastwards. The band like cloud feature remained intact until about 18 UTC with its axis gradually migrating northward.

The aerosol-cloud interaction modelling for this case is presented case in two parts. First, described in this paper, the simula-

tions with a novel microphysics scheme are evaluated against observational data from the COPE campaign and the physical mechanism of the aerosol-cloud interactions. Part 2 extends this analysis to all simulations in the meteorological ensemble and addresses questions related to the observability of the aerosol induced changes in cloud properties. For part 1, the next section provides a detailed description of our modelling system, the microphysics scheme and the observations used for evaluation. In section 3 the modelled cloud field is compared to observations, aerosol-induced changes are described in section 4, and

the mechanisms responsible for these changes are discussed in section 5. The results are finally discussed and summarised in section 6.





## 2 Data and Methods

### 2.1 Model set-up

The Unified Model (UM, version 10.3) is used for the simulations presented in this study. The UM is developed by the Met Office for operational forecasting over the UK and a range of different geographical locations (e.g., New Zealand and Aus-

5 tralia). A global model run (UM vn8.5, GA6 configuration, N512 resolution, Walters et al. (2017)) starting from the Met Office operational analysis for 18 UTC 02. 08. 2013 provides the initial and boundary conditions for a regional simulation with a grid-spacing of 1 km (500 by 500 grid-points) over the southwestern peninsula of the UK (Fig. 1a). Simulations with a grid-spacing of 250 m (900 by 600 grid-points) are nested within the 1 km simulations. A stretched vertical coordinate system is used with 120 vertical levels between the surface and 40 km altitude. The level spacing is about $40\,\mathrm{m}$ in the boundary layer

and $500\,\mathrm{m}$ at $5\,\mathrm{km}$ altitude. The nested simulations are started at 00 UTC 03. 08. 2013 and run for 24 h. Only results from the highest resolution simulations (grid spacing $\Delta x = 250$ m) will be discussed in this article.

Moisture conservation in the regional model domain is enforced using the scheme by Aranami et al. (2014) and Aranami et al. (2015). Conservation of moisture is an important physical constraint and impacts the precipitation response to aerosol perturbations (not shown). Mass conservation is also a requirement for the condensate budget analysis conducted in section 5.

In addition to the standard model code we replaced the operational microphysics with the newly developed Cloud-AeroSol Interacting Microphysics (CASIM) module, which is described in more detail in section 2.2. The CASIM module provides options for one- or two-way coupling between aerosol and cloud properties and simulations are performed in both modes.

Aerosol initial and boundary conditions are prescribed based on aerosol size distributions derived from aircraft observations (see section 2.3). A profile of aerosol mass and number densities has been derived based on aircraft observations taken from

20 a below cloud base flight leg carried out in the morning and from various cloud free flight segments at higher altitude: Mass mixing ratio and number concentration that are constant with height are used for each aerosol mode in the boundary layer and the free troposphere (Fig. 1b, Table 2). A linear transition between the two concentrations is assumed in a 500 m vertical slice centred at the mean boundary layer top of the model (1.15 km). $10\,\%$ of the observed accumulation mode aerosol is considered to be insoluble and acts as ice-nucleating particles. No surface sources of aerosol have been included. Neglecting

surface source is not expected to have a large impact on the simulations, because: (i) the chosen aerosol profiles are based on observational data over the peninsula and therefore are representative of the environment in which the clouds form; and (ii), the residence time of air in the model domain is only several hours. According to the National Atmospheric Emissions Inventory data for 2014 (NAEI, 2014) the average PM2.5 (PM1) emission flux over the model domain is $5.30\cdot 10^{-12}\,\mathrm{kg\,m^{-2}s^{-1}}$ ($2.75\cdot 10^{-12}\,\mathrm{kg\,m^{-2}s^{-1}}$). Assuming the emitted aerosol is evenly distributed over the boundary layer and with a mean flow

velocity of $7.5\,\mathrm{m\,s^{-1}}$, the resulting change in aerosol mass mixing ratio is $1.75\cdot 10^{-10}\,\mathrm{kg\,kg^{-1}}$ ($9.1\cdot 10^{-11}\,\mathrm{kg\,kg^{-1}}$). This corresponds to about $1\,\%$ ($0.5\,\%$) of the total aerosol mass or $5\,\%$ ($2\,\%$) of the accumulation mode aerosol mass in the boundary conditions. The aerosols replenishment by advection is sufficient to avoid a too strong depletion by scavenging inside the domain (see section 4.1).

For the perturbed aerosol simulations, the aircraft derived aerosol profiles are multiplied by factors of 10 and 0.1, while conserv-





ing the mean diameter of each mode. These simulations are referred to as "high" aerosol and "low" aerosol runs, respectively. For additional tests on the thermodynamic limitations of the precipitation response, simulations with aerosol concentrations increased by a factor 30 were conducted ("very high"). The simulation with the unperturbed aircraft derived profile is named "standard" aerosol run.

The regional simulations are run without a convection parameterisation. Subgrid-scale variability of relative humidity is not considered for droplet activation and condensation. Boundary layer processes, including surface fluxes of moisture and heat, are parameterised with the blended boundary layer scheme (Lock et al., 2015) and sub-grid scale turbulent processes are represented with a 3D Smagorinsky-type turbulence scheme (Halliwell, 2015; Stratton et al., 2015). Radar reflectivity has been calculated from the model fields assuming Rayleigh scattering only and neglecting extinction. Phase mixtures of hydrometeors,

i.e., partly liquid particles, are not considered.

## 2.2 CASIM microphysics and aerosol processing

For the representation of cloud microphysical processes and their interaction with the aerosol environment we use the newly developed CASIM module for the Unified Model (Shipway and Hill, 2012; Hill et al., 2015; Grosvenor et al., 2016). The CASIM module is a double-moment, five hydrometeor class microphysics scheme. The hydrometeor size distribution for each

15 category is described by a gamma distribution, of which the first two moments, i.e., mass mixing ratio and number concentration, are prognostic variables. In addition a fixed density, diameter-mass and diameter-fall speed relations are assumed for all categories. The simulated precipitation rate and reflectivity distributions are particularly sensitive to the assumed graupel density and diameter-fall speed relation. We have chosen to use the diameter-fall speed relation for medium density graupel from Locatelli and Hobbs (1974) with a graupel density of $250 \ \mathrm{kg m^{-3}}$, since this results in the closest agreement between modelled

reflectivity and surface precipitation rates and those derived from radar observations. Details on the assumed mass-diameter and diameter-fall speed relation, hydrometeor shapes and size distributions are provided in Table 1. Represented transfer rates between the different hydrometeor categories and the water vapour include droplet activation (Abdul-Razzak et al., 1998; Abdul-Razzak and Ghan, 2000), condensation (using saturation adjustment), primary ice formation from cloud droplets (De-Mott et al., 2010), freezing of rain drops (Bigg, 1953), secondary ice formation from rime-splintering in the Hallet-Mossop

temperature zone, vapour deposition, evaporation and sublimation, collision-coalescence between all hydrometeor categories and sedimentation of all hydrometeor categories except cloud droplets.

Aerosols are represented by three soluble modes and one in-soluble mode, each of which is described by a log-normal distribution with a prescribed width (Table 2) and two prognostic variables (mass mixing ratio and number density). The chemical and physical particle properties (density, solubility, etc.) are prescribed for each mode separately. The aerosol fields are initialised

from a spatially homogeneous aerosol profile, which is also used for the lateral boundary conditions throughout the simulations. The aerosol field is subject to advection.

For the aerosol-cloud interaction two different modes are available within CASIM: (i) one-way coupling of aerosols and cloud properties (passive mode), and (ii) two-way interaction between aerosols and clouds (processing mode). In the passive mode, aerosol fields are considered in the droplet activation and primary ice nucleation, but the aerosol fields are not modified by any





cloud microphysical process. In the processing mode, aerosol fields are modified consistently with the cloud microphysical processes. The interstitial aerosols are depleted by nucleation scavenging (droplet activation and primary ice nucleation). Impaction scavenging is currently not represented. Previous work suggests that in-cloud scavenging is the dominant wet aerosol removal processes and that impaction scavenging is only important below cloud-base (e.g., Flossmann et al., 1985; Yang et al.,

2015). Therefore omitting impaction scavenging should have no major implications for the present study. Droplet activation takes into account the different soluble aerosol modes according to Abdul-Razzak and Ghan (2000), while a small constant fraction of the insoluble aerosol mode is activated in water supersaturated conditions. For the ice nucleation all insoluble aerosols are used (not separating between interstitial and CCN activated insoluble aerosol) for the computation of ice crystal number concentrations to be consistent with the formulation in DeMott et al. (2010). Upon evaporation or sublimation of hy-

drometeors, the interstitial aerosols are replenished. For this purpose additional tracers for the soluble and insoluble aerosol mass and insoluble aerosol number in liquid and frozen hydrometeors are included. These tracers are subject to advection and sedimentation fluxes of the respective hydrometeors. During evaporation and sublimation, aerosols are released into the interstitial modes according to their diagnosed effective radius and the effective radius of the interstitial aerosol modes. One soluble aerosol particle is released for each evaporating hydrometeor. Thereby, hydrometeor collision-coalescence result in

fewer, but larger interstitial aerosols, if the hydrometeors subsequently evaporate. For insoluble aerosols, the activated number and mass are traced. The number of insoluble aerosols released upon evaporation or sublimation of the hydrometeor is identical to the number of insoluble aerosol particles in the hydrometeor. Thereby the number of insoluble aerosols is retained and is not impacted by collision-coalescence processes.

Simulations are conducted with passive and processing aerosol treatments and the impact on the model performance and the

20 simulated hydrometeor and aerosol fields are discussed in the sections 3 and 4.1, respectively.

## 2.3 Observational data from COPE

For the evaluation of the model simulations we make use of the observational data gathered from various platforms during the COPE campaign. The details of the experiment design are outlined in Blyth et al. (2015), while Leon et al. (2016) presents an overview of the campaign results. This study utilises radiosonde data and observations made with the Facility for Airborne

Atmospheric Measurements (FAAM) BAe-146 research aircraft . Radiosondes were launched at roughly two-hourly intervals from Davidstow ($50.64\,°N$, $-4.61\,°E$) between 8 UTC and 15 UTC. These provide profiles of air temperature, dewpoint temperature and wind vectors.

The aerosol initial conditions are provided by data collected by the FAAM BAe-146 aircraft. Three instruments, with overlapping size ranges were utilised. A scanning mobility particle sizer (SMPS) for measurements from $0.01 - 0.3\,\mu m$ diameter,

with a $30\,s$ scan time. A wing mounted Passive Cavity Aerosol Spectrometer Probe (PCASP) for measurements from $0.1$ to $3\,\mu m$ and a wing mounted Cloud Droplet Probe (CDP) for measurements from $2\,\mu m$ and above. The PCASP and CDP were calibrated using the methods of Rosenberg et al. (2012). The data was split into out-of-cloud straight and level legs spanning an integer number of SMPS scans. For each of these legs a three mode lognormal distribution was fitted to the data. No refractive index corrections were made to account for the potential different compositions of the aerosol. However, as we use just a single





average profile for each of the boundary layer and the free troposphere, any refractive index correction is much smaller than the variability in the measurements.

Cloud droplet number was also provided by the CDP. The sensitive sample area of this instrument was calibrated using a droplet generator and found to be approximately twice the nominal sample area in the manufacturer's specification. Vertical

wind measurements were provided by a 5-port turbulence probe on the aircraft nose, combined with Pitot tube airspeed measurements and GPS/inertial navigation unit aircraft altitude information (Petersen and Renfrew, 2009).

In addition to the campaign specific data, we use a 3D radar composite provided by the Met Office (Scovell and al Sakka, 2016). The composite data used here has a horizontal resolution of $1\,\mathrm{km}$, a vertical resolution of $500\,\mathrm{m}$ over the study area and a temporal resolution of $10\,\mathrm{min}$. In addition, we use the Radarnet 4 rainfall retrieval (Harrison et al., 2009; MetOffice, 2003),

which is also based on the operational radar network. The horizontal resolution is $1\,\mathrm{km}$ and the temporal resolution $5\,\mathrm{min}$.

## 3 Evaluation of model simulations with unperturbed aerosol fields

### 3.1 Radar reflectivity and surface precipitation

The model simulations capture the general evolution of the cloud band and the major structural features as described in the introduction (SI Fig. 1 and 2): The first larger clouds (maximum dimension of areas with column maximum reflectivity larger

than $25\,\mathrm{dBZ}$ exceeding $10\,\mathrm{km}$) appear after 11 UTC and these are organised along a line roughly in the centre of the peninsula. In the subsequent hours, clouds cluster along the convergence line with cells remaining more isolated and smaller over the western half of the peninsula and larger clusters developing further east. While the majority of clouds develop along the convergence lines, some more isolated clouds develop in other parts of the domain. A double-line feature also appears in model simulations, but is not as well defined as in the observational radar and satellite data. In agreement with observations, the mod-

elled cloud line slowly assumes a more north-westerly orientation throughout the day and starts to dissipate at around 17 UTC, i.e., slightly earlier than in the observations.

The domain-average surface precipitation rates from the operational radar network (Harrison et al., 2009; MetOffice, 2003) and the model simulations with the standard aerosol profile are compared in Fig 2a. In the model, surface precipitation occurs from about 11 UTC onwards, which coincides with the appearance of the first larger clouds along the convergence lines. The

25 radar shows some precipitation earlier originating from isolated convective cells. The model also produces isolated cells before 11 UTC, however, they remain much smaller and produce scarcely any surface precipitation, although column maximum reflectivity reaches values comparable to the observations. The smaller domain average precipitation in the model is mainly due to the overall smaller cell sizes. Domain-mean precipitation rapidly increases between 11 UTC and 13 UTC in the radar observations and between 11 UTC and 14 UTC in the simulations. During the afternoon both data-sets show consistently high

surface precipitation rates until about 17 UTC (model) and 18 UTC (observations). The cessation of precipitation is linked to the dissolution of the convergence lines. While the model captures the main evolution of the precipitation linked with the convergence lines, the peak domain-mean precipitation rate occurs about an hour later than in the radar data. The model underestimates the domain-mean surface precipitation relative to the radar derived precipitation irrespective of the chosen aerosol



treatment. The underestimation of domain average precipitation is related to a smaller extent of weakly precipitating areas ($< 4\,\mathrm{mm\,h}^{-1}$, SI Fig. 3a). High precipitation rates are overestimated by a factor 2 (SI Fig. 3a).

Quantitative estimates of precipitation rates from radar reflectivity can exhibit biases due to different assumptions of hydrometeor properties and evolution between the model and the algorithm used to derive rain rate from radar observations. Also

beam blocking can significantly affect low-level radar reflectivity. We therefore compare diagnosed radar reflectivity from the simulations with a 3D radar composite (Fig. 2b and SI Fig. 3b). The distribution of column maximum reflectivity, also known as composite reflectivity, (Fig. 2b) as well as the reflectivity close to the surface (at $750\,\mathrm{m}$ altitude, SI Fig. 3b) are in good agreement with the observed distribution: The most frequent column maximum radar reflectivity is about 5 dBZ too low in both simulations and the peak reflectivity are overestimated by about 5 dBZ in the passive aerosol simulation. The area covered

by clouds with low reflectivity ($< 10\,dBZ$) is underestimated in both simulations (Fig. 2b). The low-level radar reflectivity (at 750m, SI Fig. 3b) from the passive aerosol simulation slightly underestimates the occurrence of reflectivity values between $20-30\,\mathrm{dBZ}$. Above 35 dBZ the model agrees to within $\pm 5\,\mathrm{dBZ}$. In the simulation with aerosol processing the reflectivity in this region are also underestimated, but the occurrence of reflectivity smaller than $10\,\mathrm{dBZ}$ is overestimated. While the model underestimates the area covered by clouds with low reflectivity, the overall agreement between observed and modelled radar

reflectivity distributions is better than that seen between the radar derived rain rate and model rain rate, where the model underrepresents the medium surface precipitation rates. This suggests potential problems with the radar derived surface precipitation for medium to low precipitation rates, possibly linked to a lack of sub-cloud evaporation in the radar diagnostic (e. g., Li and Srivastava, 2001).

The 3D radar composite also provides information on the cloud structure, in particular the cloud depth. Cloud top height is de-

fined as the highest altitude with a radar reflectivity larger than $18\,\mathrm{dBZ}$. The mean cloud top height increases from about $2\,\mathrm{km}$ in the morning to $3\,\mathrm{km}$ in the afternoon in the observations and the model simulations (SI Fig. 4a). This indicates a general deepening of convective cells in correspondence to a larger convergence as the sea breeze lines establish. The maximum cloud depth in the observations only shows a small increase from about $5\,\mathrm{km}$ to about $5.5-6\,\mathrm{km}$, while maximum cloud top height in the model increases from $3.5\,\mathrm{km}$ to $5-5.5\,\mathrm{km}$ (SI Fig. 4b). The modelled mean cloud top height agrees within 200-500 m

with the cloud top height derived from the 3D radar composites (SI Fig. 4a) with a small dependency on the radar reflectivity value ($5-25\,\mathrm{dBZ}$) chosen to define cloud top. Given the vertical resolution of the radar data-set ($500\,\mathrm{m}$) and the model level spacing ($200\,\mathrm{m}$ at $5\,\mathrm{km}$) this is a reasonable agreement. The larger maximum cloud top heights in the radar observations are mainly due to higher level ice clouds likely forming outside the model domain.

## 3.2   Aircraft observations of hydrometeor number concentrations

The fraction of aerosol activated to cloud droplets is an important parameter for the aerosol effect on clouds. We therefore compare the cloud base cloud droplet number concentration (CDNC) measured by the CDP onboard the BAe-146 with the modelled cloud base droplet number concentration (Fig. 3). The aircraft data is taken from several flight legs close to cloud base (within $500\,\mathrm{m}$) sampling multiple cells along the convergence lines between 1200 UTC and 1250 UTC (red dots). In





the model, all clouds in the domain are sampled within the same time period (grey shading). Cloud base in the model is defined as the lowest vertical level in each column with cloud droplet mass larger than $1\,\mathrm{mg\,kg^{-1}}$. CDNC at cloud base generally increases with the vertical velocity in the model data and the observations, as expected. Sensitivity experiments with the aerosol size distribution used in the model suggest that a multi-mode representation of aerosols is required to match the

observed relation over the range of updraft velocities (up to $7\,\mathrm{m\,s^{-1}}$) observed and simulated at cloud base (not shown). CDNC values in the observations reach about $375\,\mathrm{cm^{-3}}$, which is most closely matched by the simulation with aerosol processing (Fig. 3b). The simulation assuming passive aerosols over-predicts maximum CDNC values by about $30\,\%$ (Fig. 3a). Aerosol processing reduces CDNC at all vertical velocities, which improves the match with the observational data.

Combining aircraft data from cloud penetrations at various altitudes throughout the day, provides some information of the

CDNC variation with height above cloud base. In the simulation with passive aerosol, mean CDNC decreases slowly with height, although CDNC values comparable to the cloud-base values are observed at all levels within the clouds (SI Fig. 5a). In the simulation with aerosol processing, CDNC decreases more rapidly above cloud base and the spread is significantly larger (SI Fig. 5b). The observational data in general suggests a decrease of maximum observed CDNC with altitude above cloud base (Taylor et al., 2016b). However, a direct comparison to the model results is not possible, since the location of the aircraft

observations relative to the updraft core is not known.

### 3.3   Radiosonde data

The overall structure of the thermodynamic profiles in the model is similar to the two hourly radiosonde data at Davidstow (SI Fig 6). The temperature agrees within $\pm 1$ K and the dewpoint temperature within $\pm 5$ K ($\pm 10$ K) below (above) a stable layer located between $5$ km and $6$ km altitude. The radiosonde data was compared to the thermodynamic profile at the grid column

closest to the release location of the radiosonde. As discussed later, the stable layer at $5-6$ km is an important feature of the thermodynamic profile for aerosol-induced changes in precipitation. This stable layer is located at the same altitude in both model and observations (SI Fig. 6). Other parameters such as the height of the $0\,^{\circ}\mathrm{C}$ level and the lifting condensation level are similar in the model and the observational data throughout the day with maximum deviations of $100$ m and $250$ m, respectively (SI Fig. 4c, d).

The comparison of the model simulations using the standard aerosol profiles compare favourably with radar and aircraft observations (air and dewpoint temperature profiles differences smaller than $\pm 1$ K and $\pm 5$ K, respectively; $0\,^{\circ}\mathrm{C}$ level, lifting condensation level and cloud top height within $\pm 250$ m; cloud-base CDNC differences smaller than $30\,\%$; domain-average precipitation and precipitation rates larger than $4\,\mathrm{mm\,h^{-1}}$ within a factor 2; radar reflectivity within $\pm 5$ dBZ for reflectivity larger than $5$ dBZ). This suggests that the model is adequately representing the important processes important for this case

providing confidence that changes predicted by aerosol perturbation experiments will be robust. Next we will look at processes affecting the aerosol fields and changes in the cloud field due to perturbations to the aerosol initial and boundary conditions.



## 4 Impact of the aerosol environment on the cloud field and surface precipitation

### 4.1 Modification of the aerosol environment by aerosol processing

The overall impact of including aerosol processing on the aerosol fields is illustrated by the Hovmöller diagrams of latitudinally averaged column-integrated aerosol number concentration in Fig. 4: Aitken and accumulation mode aerosol concentrations are reduced inside the clouds due to CCN activation. While the Aitken mode is depleted downstream of convective cells, the accumulation mode increases due to evaporative release of aerosol. The collision-coalescence processes in the cloud lead to a transfer of aerosol from the Aitken to the accumulation mode. The coarse mode aerosol is increasing in cloudy areas mainly due to sub-cloud evaporation of rain and downstream of convective cells. In the time interval between 9 UTC to 20 UTC, the Aitken mode number concentration reduces on average by 7 %, the accumulation mode increases by 15 % and the coarse mode increases by a factor 10 across the study area (6° W to 3.5° W). These estimates discount any clouds present at the downstream boundaries and take only cloud-free areas into account. The predicted changes in the coarse mode aerosol concentrations are large enough that they eventually could be identified in future aircraft campaigns by flying long north-south oriented runs up- and downstream of the convective line. Such observations would provide valuable information for the evaluation of the model representation of aerosol-cloud interactions.

A more detailed picture of the modification of the aerosol fields by the convective clouds is obtained from cross-sections along the convergence line (SI Fig. 7): The Aitken mode aerosol is depleted within cloud, below cloud, in areas of evaporated clouds (e.g., around 5.5° W) and in the convective outflow of the clouds at higher levels. The accumulation mode aerosol is also reduced inside clouds. However, below cloud-base and in areas of evaporated clouds accumulation mode aerosols are enhanced. Similar increases of the number concentration occur also at the lateral boundaries of clouds. These increases can be explained by evaporation of larger cloud droplets or rain drops that have ingested multiple CCN during growth by collision-coalescence. Evaporation occurs mainly in detrained air, sub-saturated air below cloud base and in decaying clouds. The coarse aerosol mode behaves very similarly to the accumulation mode aerosol, but the relative increases are more wide-spread and have a larger amplitude.

Aerosol processing reduces the mean (maximum) cloud-base droplet number concentration by 50 % (10 %) compared to the simulations with passive aerosols. This difference in cloud-base CDNC depends on aerosol concentration. Therefore, the perturbation by a factor 10 in aerosol concentration between the simulations translates to about a factor 7 change in mean cloud-base CDNC for passive aerosols, but only a change by about a factor 5 in simulations with aerosol processing. The impacts of aerosol processing on the cloud droplet number density are also reflected in a flatter relation between cloud droplet number densities and vertical velocity at cloud base as discussed in section 3 (Fig. 3). Besides an overall decrease of cloud droplet number concentration compared to the passive aerosol simulation (SI Fig. 5), the variability of the cloud droplet number concentration increases and the number concentration decrease more strongly towards cloud top. These changes are due to the depletion of interstitial aerosols inside the cloud in the runs with processing, which impedes secondary activation in the model. The overall reduction of the cloud droplet number concentration leads to higher rain and graupel number concentrations due to more efficient conversion of rain formation and slightly smaller ice and snow number concentrations (not shown). The vertical





variation in number concentrations is very similar for all hydrometeors except cloud droplets for the two runs.

The impact of aerosol processing on aerosol fields and hydrometeor number concentrations is qualitatively very similar for simulations with perturbed aerosol initial conditions, except for the low aerosol simulations (not shown). For this simulation the graupel number concentration displays hardly any change and the reduction of ice and snow number concentrations is

more pronounced. The impact of aerosol processing on precipitation formation and the overall cloud structure and evolution is discussed in the following sections.

## 4.2  Cloud field structure and cloud geometry

The perturbed aerosol conditions impact the cloud field structure in simulations with and without aerosol processing in a similar way (SI Fig. 8): Cloud fields from simulations with higher aerosol concentrations are more organised with larger, less

widespread and more densely packed objects. These changes are consistent throughout the simulated time period.The number of cells decreases and the mean cell size increases with larger aerosol concentrations (Fig. 5a,b). The cell number changes in all development stages of the convective line, while the change in cell size is particularly evident in the afternoon (after about 13 UTC) when the convective line is fully developed. Aerosol induced changes in cell number and size are smaller for aerosol concentrations enhanced above the standard aerosol profile compared to reduced aerosol concentrations. The transition to a

more structured cloud field in the high aerosol environment is accompanied by a small reduction in the fraction of the domain covered by cloud (Fig. 5c). The cloud fraction is defined as the proportion of the domain covered by clouds with a condensed water path larger than $1 \, \mathrm{g \, m^{-2}}$. The temporal evolution of the cloud fraction in all simulations is very similar with a gradual increase up to 14 UTC and a rapid decrease after 17 UTC. Changes in cell number and size operate such that aerosol-induced changes in cloud fraction are very small, occurring mainly between 13 UTC and 16 UTC.

In addition to the change in the horizontal cloud structure, the average cloud top height rises in high aerosol environments (Fig. 5d). Differences are rather small in the development phase of the convective line, but amount to about $100 - 200 \, \mathrm{m}$ in the afternoon. The cloud top height change is dominated by a reduction of cloud tops between $3 - 4 \, \mathrm{km}$ and an increase of cloud tops above $4.5 \, \mathrm{km}$ (SI Fig. 9a). Cloud base height variations between simulations with different aerosol profiles are much smaller (mean height $\pm 50 \, \mathrm{m}$, SI Fig. 9b). For enhanced aerosol concentrations changes in cloud top height are larger in the

aerosol processing than the passive aerosol simulation. The maximum cloud top height is restricted by a stable layer extending from about $5 - 6 \, \mathrm{km}$, which is evident in thermodynamic profiles from radiosondes and model simulations (SI Fig. 8). While cloud tops reach the stable layer in the simulations with passive aerosols under standard aerosol concentrations, clouds reach this altitude only under enhanced aerosol concentrations if aerosol processing is included. Therefore in the latter cloud deepening from the low to the high aerosol scenario is not yet limited by the thermodynamic profile. For the simulations with passive

aerosol, a similar asymmetry in the response to increases and decreases in the aerosol concentration is also evident in other variables (section 4.3 and 5). We hypothesise that this asymmetry is controlled by thermodynamic constraints on the cloud top height. This hypothesis is further discussed in section 5.



### 4.3 Surface precipitation

The accumulated surface precipitation over the time period of convective activity (9-19 UTC) depends on the chosen aerosol representation and on the aerosol profile (Fig. 6a). The accumulated precipitation is about $20\,\%$ smaller in the simulations with aerosol processing. The sensitivity to perturbed aerosol conditions is larger in the aerosol processing simulation with changes of up to $14\,\%$ relative to the standard aerosol environment, while for passive aerosol precipitation only varies by up to $4\,\%$. The larger sensitivity in the simulations with aerosol processing occurs despite a smaller perturbation in cloud-base CDNC between the different aerosol scenarios (section 4.1).

The overall precipitation response can be divided in two different phases (Fig. 6b): An early phase (9-12 UTC), for which precipitation decreases with increasing aerosols, and a later phase (12-19 UTC), for which precipitation increases with increasing aerosols. The transition from precipitation suppression to enhancement occurs with aerosol processing and passive aerosol and is related to the transition from isolated and unorganised convective clouds to on average larger and deeper cloud clusters forming along the converging sea breeze fronts from around noon (SI Fig. 1, 2).

In the early phase, precipitation continuously decreases with increasing aerosol concentrations. This agrees with parcel model arguments of a less efficient collision-coalescence in the presence of more CDNC (e.g., Twomey, 1966; Feingold et al., 2013). The precipitation response is similar in the simulations with passive aerosols and aerosol processing, but has a smaller amplitude in the latter.

In the afternoon, the accumulated precipitation increases with enhanced aerosol concentrations up to standard (high) aerosol concentrations for simulations with passive aerosols (aerosol processing) contrary to simple precipitation suppression ideas. Further enhancing the aerosol concentration does not lead to further precipitation enhancement. The physical mechanisms controlling this precipitation response are discussed in section 5.

The precipitation rate distribution is also influenced by the aerosol concentrations (SI Fig. 10a). In the passive aerosol simulations medium rain rates ($\approx 1 - 20\,\mathrm{mm\,h^{-1}}$) are more frequent and small rain rates less frequent with increasing aerosol concentrations. High precipitation rates ($> 30\,\mathrm{mm\,h^{-1}}$) occur more frequently in the simulations with the standard aerosol profile than those with perturbed aerosol profiles. If aerosol processing is included, changes in low and medium rain rates are much smaller and the probability of high rain rates ($> 30\,\mathrm{mm\,h^{-1}}$) increases with the aerosol concentration. The aerosol induced changes in accumulated surface precipitation and the precipitation rate distribution, suggest that precipitation formation is enhanced by increased aerosol concentrations in the simulation with aerosol processing and to a lesser extent in simulations with passive aerosols.

### 4.4 Condensed water content

Under enhanced aerosol conditions, precipitation formation is thought to be suppressed due to less efficient conversion of condensed water to precipitation seized hydrometeors. Consistent with this idea, the domain-integrated condensed water path increases with the aerosol concentration in the time period of main convective activity, i.e., between 12 UTC and 18 UTC (SI Fig. 11a). This increase is larger and extends to the entire simulated time period, if only hydrometeors with small sedimentation





velocities (cloud droplets, ice and snow) are taken into account (SI Fig. 11b).

Parcel model considerations suggest that a higher condensed water content is required to obtain the same precipitation rate for higher cloud droplet number concentrations due to less efficient collision-coalescence growth of hydrometeors. To investigate whether this also holds for the relation of condensate loading and precipitation in a more complex model, the distribution

of condensate loading (in the ice and cloud droplet categories) in columns with a specific precipitation rate are displayed in Fig. 7. The $90^{th}$ percentile of condensate loadings is increasing from low to high aerosol concentrations independent of the assumed aerosol-microphysics coupling. For lower percentiles, the condensed water path only increases up to the standard aerosol concentration and remains unchanged or decreases for increasing the aerosol concentrations. While these changes occur systematically over a wide range of precipitation rates, they are still relatively small compared to the spread of the

condensed water path at a given precipitation rate. Differences to the expectations from parcel models likely arise, because statistics are computed over different lifecycle stages and the cloud microphysics operate in a "vertical column", i.e., the condensed water path does not directly translate into condensed water content and falling hydrometeors affect precipitation production at lower levels in the cloud. This more complex situation likely results in a less straightforward relation between condensed water path and precipitation rate than expected from parcel models. We hypothesise that the signal is consistent with

parcel models in the highest percentile, because these correspond to updraft regions in clouds at the mature lifecycle stage, for which condensate production is compensating for losses due to precipitation production. This is consistent with very small changes in the condensed water path percentiles including all hydrometeor species, i.e., also including snow, rain and graupel (SI Fig. 12c,d).

## 5   Physical mechanism of aerosol induced changes

**5.1   Decomposition of precipitation response to changes in condensate generation and loss**

Changes in precipitation (aerosol induced, as in this study, or other) can be contextualised and interpreted by investigating the condensate budget of the considered clouds (e.g., Barstad et al., 2007; Khain and Lynn, 2009; Altaratz et al., 2014). The change in precipitation at the surface $\Delta P$ is thereby considered as the result of changes in condensate gain $\Delta G$ by condensation and deposition of water vapour and condensate loss $\Delta L$ by evaporation and sublimation. If changes in condensate gain are larger

than changes to the loss terms, surface precipitation increases and vice-versa. Furthermore, the ratio between $\Delta G$ and $\Delta L$ combined with the precipitation efficiency of the control simulation indicates whether changes in the condensate generation, i.e., changes in uplift, are driving surface precipitation responses or whether changes in precipitation efficiency, i.e., more or less efficient conversion of condensate to precipitation, are involved as well (see Appendix). The analysis of the condensate budget provides a close link between cloud microphysics, cloud dynamics and the cloud environment, since it is intrinsically

linked to latent heating, the condensate distribution within the cloud and the different timescales of cloud microphysics and dynamics (e.g., Stevens and Seifert, 2008; Miltenberger et al., 2015).

The condensate budget approach requires reasonable mass conservation properties of the underlying numerical model and a closed mass budget over the cloud system lifetime. The first is ensured in our simulation by using the Aranami et al. (2014)





and Aranami et al. (2015) approach to enforce moisture conservation in the regional model domain. The domain used in our simulation does not cover the entire length of the convergence lines and therefore the condensate mass budget cannot be closed over the entire life cycle of all clouds. Most cloud systems do, however, reach their mature stage before being advected out of the domain. We diagnose the advected condensate amount at the domain edge from meteorological fields at 10 min resolution.

Fig. 8a shows that the inclusion of the advective terms has a small impact on the changes in gain and loss terms (compare open and filled symbols). Advective terms will therefore be ignored for the rest of the analysis.

The condensate budget terms are calculated by integrating condensation, evaporation, deposition and sublimation rates over the model domain and the time period of convective activity, i.e., between 9 UTC and 20 UTC. Changes in the condensate generation $\Delta G$ and loss terms $\Delta L$ relative to simulations with the standard aerosol profiles are shown in Fig. 8a. All points

above the one-to-one line correspond to simulations, for which the condensate gain changes less than the condensate loss and which accordingly have reduced surface precipitation. Points below the one-to-one line portray simulations with enhanced precipitation. The sign of condensate gain is correlated with the sign of the aerosol perturbation, i.e., an increased condensate gain for higher aerosol concentrations. The magnitude of $\Delta L$ roughly scales with $\Delta G$ except for the simulation with low aerosol concentration and aerosol processing. However, the absolute magnitude of $\Delta G$ is always larger than $\Delta L$. The only exception

are the simulations with high and very high aerosol concentrations and passive aerosols, for which $\Delta L$ approximately equals $\Delta G$. This behaviour suggests that the aerosol induced changes in surface precipitation are primarily controlled by changes in the condensation and deposition terms.

To investigate whether changes in the condensation generation alone or accompanying changes in the precipitation efficiency are more important for the surface precipitation response, we decompose the precipitation response according to equation A2

(Appendix) in Fig. 8a and Table 4. Precipitation efficiency PE is defined here as the ratio of domain integrated time accumulated surface precipitation and condensate gain and quantifies the efficiency of cloud microphysical processes to convert condensate to surface precipitation. Precipitation efficiency changes by up to 2 % between simulations with different aerosol concentrations. Cloud base precipitation efficiency, i.e., discounting sub-cloud evaporation of rain, is overall about 10 % larger, but behaves in a similar way. PE changes are generally larger in simulations with aerosol processing than in those with passive

aerosols. Simulations for which changes in $\Delta G$ alone are more important than changes in PE, fall in Fig. 8a in the area between the one-to-one line and the blue dashed lines. The majority of the simulations are located close to the dashed lines indicating changes in G are of comparable importance to changes in PE. In general changes in PE are more important in the simulations with aerosol processing.

The terms in the condensate budget can be further split according to the phase state of the involved hydrometeors, i.e., into

condensation/evaporation and deposition/sublimation. The changes in these four terms, again relative to the simulation with the standard aerosol profile, are shown in Fig. 8b. Absolute changes in condensation and evaporation (filled symbols) are generally larger than changes in deposition and sublimation (open symbols). The only exception is the simulation with aerosol processing and a high aerosol concentration, for which both are of similar magnitude. The changes in terms involving liquid and solid hydrometeors have in general the same sign, i.e., if condensation increases, deposition does so as well. $\Delta L$ is very small for

solid phase hydrometeors suggesting (i) that the contribution of solid phase hydrometeors to total precipitation increases with





aerosol and (ii) that detrainment and subsequent sublimation of solid phase hydrometeors does not significantly increase with aerosol concentrations for simulations with passive aerosol.

From the analysis of the condensate budget so far we can hence conclude that the aerosol-induced changes in surface precipitation are driven by changes in the condensation and evaporation, while changes in sublimation and deposition are of minor

importance For most cases changes in the total condensate gain are of similar importance to changes in precipitation efficiency. Changes in the loss terms (and therefore the precipitation efficiency) are important to understand differences between passive aerosol and aerosol processing simulations and the simulations with very high aerosol concentrations. For simulations with reduced (enhanced) aerosol concentrations, condensation gain is smaller (larger) than in the control case as is the accumulated surface precipitation (Fig. 8a, 6a). The only exceptions are simulations with very high (and, for passive aerosols, high)

aerosol concentrations, in which precipitation does not increase further. The mechanisms driving changes in condensate gain are investigated in more detail in the next section.

## 5.2 Aerosol impact on convective core and stratiform regions

For a closer analysis of the driving factors behind $\Delta G$ and $\Delta L$, the cloud field is decomposed into regions with different updraft strength. Cloudy columns are stratified according to the column maximum in-cloud vertical velocity at each grid point

($w_{max}$). $\Delta G$ for the conditionally sampled areas of the domain is shown in Fig. 9a. Condensation changes are dominated by regions with large $w_{max}$, while smaller changes of opposite sign occur in weaker updraft regions. In contrast, weak updraft regions contribute most to changes in deposition. These modifications go along with changes in the vertical extent and area covered by the updraft regions (Fig. 9b, SI Fig. 13a). Updraft cores deepen by about $100 - 150\,\mathrm{m}$ for each factor 10 increase of aerosol, while the areal extent increases by about $25\,\%$. These changes in the updraft geometry do contribute to the differences

in column integrated condensate generation between simulations. However, they do not fully explain them, since the averaged condensation rate also increases with aerosol concentration (SI Fig. 13b). These changes are consistent between simulations with and without aerosol processing.

Different responses to aerosol perturbations are expected in the convective core regions and the more stratiform regions of the cells. Due to the change in behaviour of $\Delta G$, updraft width and depth between $3\,\mathrm{m\,s^{-1}}$ and $4\,\mathrm{m\,s^{-1}}$, updraft regions are

25 defined by $w_{max} > 3\,\mathrm{m\,s^{-1}}$ and the more stratiform regions by $w_{max} = [0,3]\,\mathrm{m\,s^{-1}}$. Average profiles of kinetic energy, latent heating rates and total condensed water for the two regions are shown in Fig. 10.

In the convective core regions the kinetic energy and the latent heat release increases with increasing aerosol concentrations (Fig. 10b, d). Both of these variables peak in the warm-phase part of the cloud, with the peak in kinetic energy occurring about $1\,\mathrm{km}$ above the peak in latent heat release. The maximum in both variables shifts to higher altitudes with increasing aerosol

concentrations. Latent heat release above the $0\,^\circ\mathrm{C}$ level increases slightly for higher aerosol scenarios, but the changes are very small compared to those below the $0\,^\circ\mathrm{C}$ level. The condensate mass in the lower parts of the cloud decreases with increasing aerosol, while it increases above the $0\,^\circ\mathrm{C}$ level (Fig. 10f). The altitude of the maximum condensate loading also shifts to higher altitudes for higher aerosol loadings. The generally small changes above the $0\,^\circ\mathrm{C}$-line indicate that the precipitation enhancement is mainly a result of changes in the warm-phase part of the cloud, as suggested by the analysis in section 5.1.



While energy released from phase transitions in the lower part of the cloud increases, the vertical velocity is almost unaltered as is the cloud base temperature ($\Delta T < 0.1\,^{\circ}C$). Therefore the higher condensation rates can only be explained by less dry air being mixed into the high updraft regions for increasing aerosol conditions. This hypothesis is corroborated by the decreasing contribution of evaporation in simulations with higher aerosol concentrations (SI Fig. 14). Less mixing with dry air is consis-

5 tent with on-average larger cells and an increasing stratiform area (Fig. 5b, 9b). The stronger latent heating from convection as well as the less strong mixing with low kinetic energy air masses (due to a wider updraft region) contributes to the higher vertical velocities aloft. The larger vertical motion promotes the upward transport of condensate. The higher condensate amounts towards cloud top are also supported by slower conversion rates of cloud condensate into rain (SI Fig. 16). The more efficient rain production in low aerosol conditions also explains the larger cloud condensate mass close to cloud base, which is mainly

a result of the sedimentation of rain (SI Fig. 16).

In the stratiform region, the kinetic energy in the lower parts of the clouds is not affected by modified aerosol concentrations, while it increases in the higher parts of the clouds for higher aerosol concentrations (Fig. 10a). With increasing aerosol concentrations, the latent heat release close to cloud base slightly increases, while it becomes more negative in the upper part of the clouds (Fig. 10c). The condensed water content also shows small increase close to cloud base, little change up to $2\,km$ and

15 a strong increase aloft (Fig. 10e). The changes in the upper parts of the clouds are due to a larger horizontal transport of condensed water into the stratiform regions of the clouds. This is caused by a stronger divergence in the upper parts of the clouds in direct consequence of a higher vertical flux in the convective core region. Secondly, the higher condensate content in the upper parts of the clouds enables lateral mixing to broaden the cloud. In contrast for low condensate content, lateral mixing leads to an evaporation of the cloud. The broadening of the clouds by larger transport into the stratiform regions is consistent with the

20 overall larger cloud size in the scenarios with higher aerosol concentrations (Fig. 5b). A further impact of the lateral mixing is that the convective core regions are less affected by entrainment of cloud free area, which supports the formation of wider and deeper regions with high updrafts. The increase in condensate loading in the stratiform region is particularly pronounced for simulations in which vertical growth of the clouds is prohibited by the stable layer aloft (high aerosol scenario with passive aerosols). This enhanced export of condensate to the stratiform region together with a less efficient rain and graupel production

(SI Fig. 16) explain the absence of a further precipitation increase in these simulations compared to those with the standard aerosol profile.

The discussed changes in cloud dynamics and microphysics are consistent between the simulations with passive aerosol and aerosol processing. The difference between the passive and processing treatment of aerosols can be understood based on the overall lower CDNC in the aerosol processing simulations with the same physical mechanism.

### 30  5.3  Transition from precipitation suppression to enhancement

The discussion of precipitation changes in the previous two sections focused on the precipitation enhancement with increasing aerosol concentrations during the period of main convective activity (12-20 UTC). However, in the earlier period with scattered and shallow convection, precipitation was suppressed by higher aerosol concentrations (section 4.3). In the morning, most clouds are less deep and the cloud top height shows only small changes with the different aerosol scenarios (Fig. 5d). Due to





the generally lower cloud tops, the clouds are predominantly warm-phase with very small ice and snow contributions close to cloud top. Consistent with the small change in cloud top height, the condensate gain displays only minor changes. The changes in the condensate loss are more significant and in contrast to the clouds developing later dominate the condensate budget (Fig. 11). Systematic changes in cell sizes are much smaller than later on (Fig. 5b). The lack of ice-phase species and the lower

wind-speeds at cloud top limit the lateral transport of condensate into the area surrounding the updraft core and thereby prevent cells from growing larger. Accordingly the main control on the precipitation formation is the efficiency of cloud microphysical processes in producing precipitation sized hydrometeors. This efficiency rapidly increases with decreasing CDNC and hence aerosol concentrations.

In the later period, deeper clouds develop (Fig. 5a) due to the additional forcing from the converging sea-breeze fronts. Max-

10 imum in-cloud vertical velocities often exceed $3\,\mathrm{m\,s^{-1}}$, a larger mixed-phase region develops and the cloud depth and cell width become more sensitive to the ambient aerosol concentrations. The differences in accumulated precipitation changes are mainly due to changes in the higher precipitation rates (precipitation rates larger than $4\,\mathrm{mm\,h^{-1}}$ account for $80\,\%$ of $\Delta P$, SI Fig. 10b). Changes in precipitation rate are mainly occurring in the regions with high vertical velocities (SI Fig. 13c). While the mean profiles show smaller rain and graupel mass mixing ratios with increasing aerosol concentration inside the cloud,

the rain mass mixing ratio in the below cloud region does not vary strongly. In contrast the frequency of high graupel water paths increases with increasing aerosol concentrations (SI Fig. 16). The composite is averaged over the number of hydrometeor containing columns at each level, i.e., an increase in peak graupel and rain mixing ratios will be masked by the more numerous instances without significant graupel and rain due to the longer timescales needed for the formation of precipitation sized hydrometeors. Hence the increasing maximum precipitation rates under higher aerosol concentration are a consequence

of the higher condensate content in the convective core, which ultimately leads to higher rain rates although later in the cloud lifecycle. The overall decrease in the highest aerosol scenarios is caused by the balance of increased lateral flux of condensate into the more stratiform regions and the even further reduced formation efficiency of rain and graupel.

The aerosol-induced changes in cloud structure, lifecycle and precipitation formation for the phase of organised convection are summarised in Fig. 12. The right (left) column corresponds to clouds, for which the vertical development is (not) limited

by a stable layer aloft. The upper panels depict the control scenario, while the lower panels indicate the cloud evolution under increased aerosol conditions.

## 6   Discussion and Conclusions

Aerosol-cloud interactions are investigated for mixed-phase convective clouds developing along a sea-breeze convergence zone over the southwestern peninsula of the UK. High-resolution ($\Delta\mathrm{x} = 250\,\mathrm{m}$) simulations with the Unified Model have

30 been conducted with a newly developed cloud microphysics scheme (CASIM), which can represent the modification of the aerosol environment by cloud microphysical processes. Evaluation of the model simulations with observations from the COPE campaign suggest a good model performance in terms of the thermodynamic profiles, cloud structure, cloud microphysical and radar reflectivity structure. The good agreement with low-level radar reflectivity, but larger difference in surface precipitation





rate, suggest potential issues with the radar-retrieved surface precipitation rates.

A novel aspect of CASIM is the representation of the modification of aerosol environment by cloud microphysical processes in a numerical weather prediction framework. Including this feedback has a largely positive impact on the model performance in terms of cloud-base cloud droplet number density and reflectivity, but leads to a stronger underestimation of domain-average

surface precipitation and smaller cell sizes. The most notable changes are in the cloud droplet number concentrations and its spatial distribution, which also results in changes in cloud geometry and the amplitude, but not the mechanism of the response to aerosol perturbations:

1. Aerosol processing reduces cloud base CDNC, results in a more rapid decrease of CDNC with altitude and increases the spread of CDNC values at each altitude.

2. Changes in precipitation in the first phase of unorganised convection are smaller, when aerosol processing is included, compared to simulations with passive aerosols.

3. In the second phase with more organised convection, changes in precipitation are larger, when aerosol processing is included, and changes in precipitation efficiency become more dominant. Changes appear larger due to overall less deep clouds with aerosol processing for a given aerosol concentration and cloud deepening becomes thermodynamically

limited only in higher aerosol environments.

The two-way interaction between clouds and aerosols is an important feedback mechanism, which may impact the magnitude of aerosol-induced changes in clouds and is one source for co-variability between cloud and aerosol fields. The latter factor is particularly important on larger spatial and temporal scales, which typically cannot be represented in very high-resolution simulations with very detailed microphysics. Hence, one-way ,i.e., using aerosol for CCN activation, or two.way coupling,

i.e., changing the aerosol fields according to microphysical process, between aerosol and cloud fields has been implemented in some numerical weather prediction models with bulk microphysics schemes (e.g., COSMO-ART, Vogel et al. (2009); COSMO-MUSCAT Dipu et al. (2017); or WRF-CHEM Fast et al. (2006)). Given the recent development of these modelling systems only a limited number of studies on the sensitivity of aerosol-induced changes in clouds are available, which have predominantly focused on stratocumulus clouds and aerosol processing. In this work, we have shown the importance of aerosol-processing in

orographically forced convection.

Perturbations to the aerosol initial and boundary conditions (modifications by factors 0.1, 10 and 30) cause distinctive changes in the cloud microphysical properties, geometry and precipitation production. These changes are summarised in Fig. 12. Key aspects are:

1. Precipitation suppression under high aerosol conditions transitions to precipitation enhancement, when the clouds or-

ganise and grow deeper.

2. Changes in precipitation are mainly a result of modified condensate generation in the warm-phase part of the clouds and changes in precipitation efficiency. The latter are more important for simulations with aerosol processing.



3. An increase of vertical velocities in the convective core regions and of condensate production, i.e., convective invigoration, occurs for enhanced aerosol concentrations.

4. The translation of the convective invigoration to precipitation enhancement is limited by thermodynamic constraints on the cloud depth.

5. The changes in the cloud field structure and mixing are important for the response to aerosol perturbations.

6. Aerosol perturbations modify the cell number and sizes, but have little impact on the domain cloud fraction.

The change in precipitation response with the transition from shallow, unorganised to deeper and more organised convection is in line with previous results from individual simulations in different cloud regimes as summarised for example by Khain (2009). Different precipitation responses for convective and more stratiform precipitation have also been documented in the large domain simulations of tropical convection by Lee and Feingold (2010). Previous studies on aerosol-induced changes in precipitation formation in deep convective clouds mainly focussed on changes in latent heating in the mixed-phase part of the clouds (e.g., Rosenfeld et al., 2008; Lebo and Seinfeld, 2011), whereas our simulations suggest that the precipitation response is mainly driven by changes in latent heating below the $0\,^{\circ}$C level. A potential reason for this difference may be that clouds in the COPE case have a less deep mixed-phase section compared to deep convective clouds in previous studies, which typically reach the tropopause. We hypothesis that changes in latent heating rates are related to changes in cloud field structure. In terms of changes in cloud structure, most previous studies have focussed on changes in cloud top height or cloud depth (e.g., Koren et al., 2005; Stevens and Feingold, 2009; Morrison and Grabowski, 2011), but changes in lateral cloud structure have received less attention. Most previous studies used either small-domain, high-resolution simulations, which are unable to represent large changes in cloud field structure, or larger-domain, but coarser resolution simulations, which lack a representation of updraft dynamics. In the present study, spatial resolution is high enough to at least partly resolve updraft dynamics and the domain is large enough to represent cloud-cloud interactions and allow for changes in cloud field structure. The small changes in cloud fraction, but major changes in cell number and area support the idea of the importance of changes in cloud field structure and related compensating mechanisms as previously suggested for example by Stevens and Feingold (2009).

Despite the fairly high resolution of the presented simulations, there are some questions to the representation of lateral mixing in the model simulations. Numerical weather prediction models have known issues with reproducing observed cell size distributions and modelled cell size distributions do not converge in simulations with increasing spatial resolution (e.g., Stein et al., 2014, 2015; Hanley et al., 2015). These problems have been at least partly attributed to the representation of the lateral mixing and parameter settings therein (Stein et al., 2015; Hanley et al., 2015). Future studies should investigate the sensitivity of the aerosol induced changes in cloud field structure to the representation of lateral mixing or test whether similar changes occur in models better resolving lateral mixing (LES simulations). Another caveat to the presented simulations is the use of saturation adjustment in the CASIM microphysics model. Lebo and Seinfeld (2011) have found significant differences in the magnitude of aerosol-induced changes in latent heating in the mixed-phase part of the cloud depending on the representation of supersaturations. Changes in latent heating were found to be much smaller if saturation adjustment was used. The representation of mixed-phase cloud microphysics in models has a number of other uncertainties, both of parametric and structural





nature. These include, but are not limited, to the representation of primary and secondary ice nucleation, drop freezing, rimed particle density, and diameter-fall speed relations (e.g., Morrison, 2012; Johnson et al., 2015; Huang et al., 2017). Most of these uncertainties in the microphysics are expected to influence the precipitation efficiency. However, given that changes in condensate generation play an important role in the studied clouds, it can be speculated that these changes may have an impact

on the overall precipitation, but not on the overall mechanism of precipitation response.

In the second part of this study, we will investigate whether the aerosol-induced changes in cloud structure and precipitation discussed here are consistent across an initial condition ensemble (perturbations to meteorological variables) for the same case. Further, the aerosol-induced changes will be put into the context of changes induced by small modification of the meteorological initial conditions. This provides insight into the detectability of aerosol-cloud interactions in observational data and the

10 demands on observational data to enable a detection of aerosol-induced changes.

*Data availability.* Model data is stored on the tape archive provided by JASMIN (http://www.jasmin. ac.uk/) service. Data access to Met Office data via JASMIN is described at http://www.ceda.ac.uk/blog/access-to-the-met-office-mass-archive-on-jasmin-goes-live/.

## Appendix A: Appendix

In a mass-conserving system, where changes to condensate storage are negligible, the surface precipitation P equals the differ-

15 ence between condensate generation G (condensation and deposition) and condensate loss L (evaporation and sublimation):

$$P = G - L \qquad (A1)$$

The condensate generation is mainly determined by the cloud dynamics, i.e., uplift in saturated conditions, and to a smaller extend the efficiency with which the generated supersaturation is depleted by transfer to the condensed phase. The condensate loss is determined by the efficiency of microphysical processes to convert condensate to surface precipitation and the time-

20 scale available for this conversion, i.e., the residence time of any infinitely small air parcels in (super-)saturated conditions. Accordingly the change in precipitation between two different cases is the result of changes in the generation and loss terms: If the change in loss are larger than those in the generation term, precipitation will decrease and vice-versa. A convenient way to display this analysis is therefore a plot of $\Delta G$ against $\Delta L$ (Fig. A1).

This analysis can be extended to address the question whether a specific change in surface precipitation is dominated by a

25 change in the generation term or a change in the conversion efficiency. For this purpose, the precipitation efficiency PE is used, which is defined as the ratio of surface precipitation to condensate generation. The change in surface precipitation can be decomposed according to:

$$\begin{aligned}
\Delta P = P_{ctr} - P_{per} &= G_{ctr} PE_{ctr} - G_{per} PE_{per} \\
&= G_{ctr} PE_{ctr} - G_{per} PE_{per} - G_{per} PE_{ctr} + G_{per} PE_{ctr} \\
&= PE_{ctr} \Delta G + G_{per} \Delta PE \qquad (A2)
\end{aligned}$$





The first term on the right side of the equation quantifies the contribution of a change in generation and the second term those of an altered precipitation efficiency. The conditions, for which the change in condensate generation dominate, are accordingly:

$$|PE_{ctr}\Delta G| > |G_{per}\Delta PE| = |G_{per}\frac{P_{crt}}{G_{ctr}} - P_{per}|$$
$$= |G_{per}\frac{P_{crt}}{G_{ctr}} - P_{per}| = |-\frac{G_{per}}{G_{ctr}}L_{ctr} + L_{per}|$$
$$= |-\Delta L - \Delta G(1 - PE_{ctr})| \tag{A3}$$

These conditions are met by the following combinations of $\Delta G$ and $\Delta L$:

$$1. \Delta G > 0 \quad \& \quad \Delta L < \Delta G \quad \& \quad \Delta L > \Delta G(1 - 2 \cdot PE_{ctr}) \tag{A4}$$

$$2. \Delta G < 0 \quad \& \quad \Delta L > \Delta G \quad \& \quad \Delta L < \Delta G(1 - 2 \cdot PE_{ctr}) \tag{A5}$$

The respective areas in the $\Delta G - \Delta L$ are illustrated in Fig. A1.

*Author contributions.* All authors contributed to the development of the concepts and ideas presented in this paper. B. J. Shipway developed the CASIM microphysics code. A. A. Hill, J. M. Wilkinson, P. R. Field and A. K. Miltenberger contributed to the further development of the CASIM code. A. K. Miltenberger, P. Rosenberg, and P. R. Field helped set up the model runs. R. Scovell provided the 3D radar composite. P. Rosenberg compiled and analysed the aircraft data set. A. M. Blyth provided expertise on the observational data sets and the observational campaign. A. K. Miltenberger performed the model simulations and model analysis, and wrote the majority of the manuscript, along with input and comments from all co-authors.

*Competing interests.* The authors declare that they have no conflict of interest.

*Acknowledgements.* We thank the COPE research team for collecting observational data and in particular John Taylor from the University of Manchester for discussion related to the cloud droplet data. Further, we acknowledge use of the MONSooN system, a collaborative facility supplied under the Joint Weather and Climate Research Programme, a strategic partnership between the Met Office and the Natural Environment Research Council. Further we acknowledge JASMIN storage facilities (doi : 10.1109/BigData.2013.6691556), FAAM, CEDA, BADC and the Radarnet at the Met Office team for providing data. The University of Leeds is acknowledged for providing funds for this study.



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





(a)                                                                          (b)

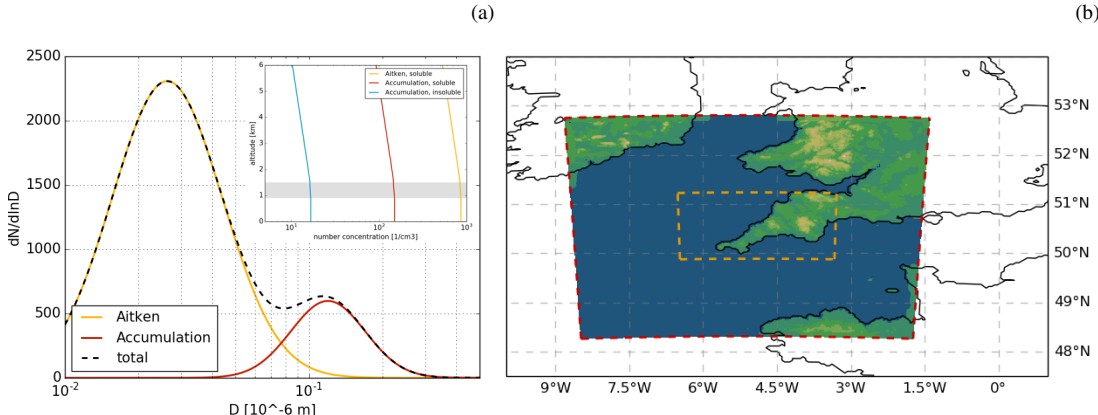

**Figure 1.** (a) Aerosol size distribution used for initialising the simulations in the boundary layer: Aitken model (orange line), accumulation mode (red line) and the sum of both (black dashed line). The vertical profiles of the aerosol number density is shown in the inset. The grey horizontal bar showing the 500 m interval around the mean height of the boundary layer, across which a linear decrease of boundary layer aerosol concentrations to free tropospheric concentrations is prescribed. (b) Regional model domains used for the simulations: the red dashed line corresponds to the domain boundary used for the 1 km simulations and the orange dashed line to the one for the 250 m simulation.





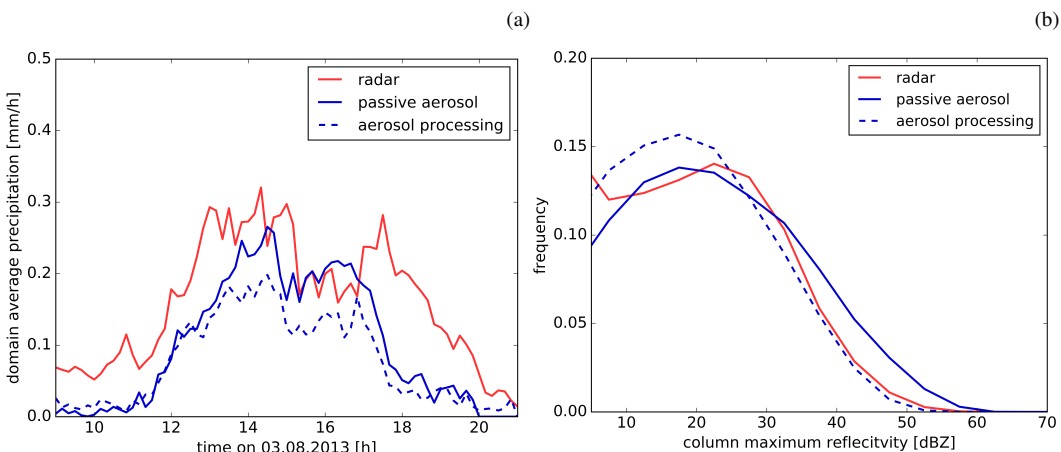

**Figure 2.** Comparison of the (a) domain mean surface precipitation rate and (b) distribution of column maximum radar reflectivity from model simulations with the standard aerosol profile (blue) and radar observations (red). The solid line shows results from the simulation with passive aerosols and the dashed line from the simulation with aerosol processing. Simulated precipitation rates and radar reflectivity have been coarse-grained to the spatial resolution of the radar observations (1 km horizontal and 500 m vertical).

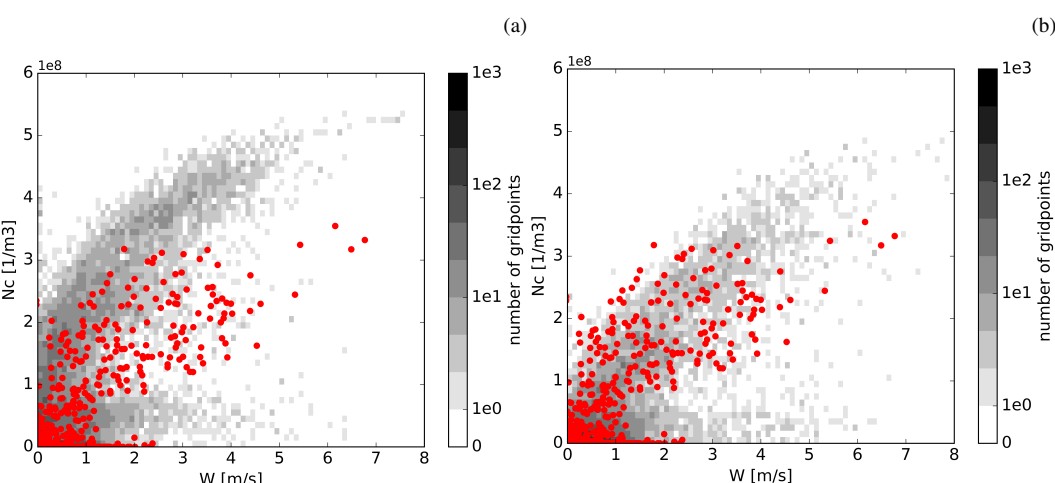

**Figure 3.** Cloud base cloud droplet number density as a function of vertical velocities from aircraft data (red symbols) and model data (grey shading). Aircraft observations include data collected during low-level flight legs close to cloud base ($\bar{z} = 1160$ m, 1200 UTC to 1250 UTC). CDP measurements are used for the cloud droplet number concentrations and (AIMMS)-20 measurements for the vertical velocity. Cloud base cloud droplet number density and vertical velocity in the model is retrieved from the lowest model level with cloud droplet mass larger than $1 \, \mathrm{mg \, kg^{-1}}$ from the entire domain between 12 UTC and 13 UTC.



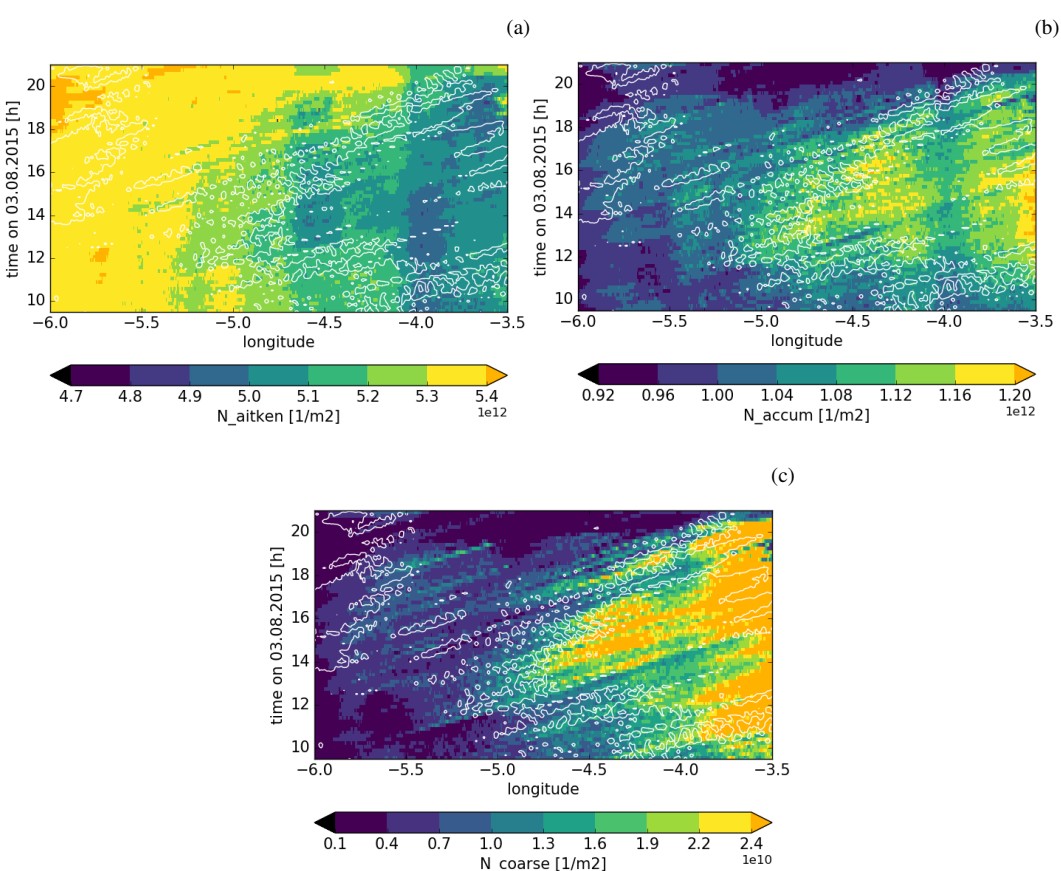

**Figure 4.** Hovmöller diagrams of latitudinally averaged column integrated aerosol number density from the simulation with aerosol processing and the standard aerosol profile for the Aitken (a), accumulation (b) and coarse mode (c). The white contour indicates areas where the condensed water path is larger than $0.1\,\mathrm{kg\,m^{-2}}$.




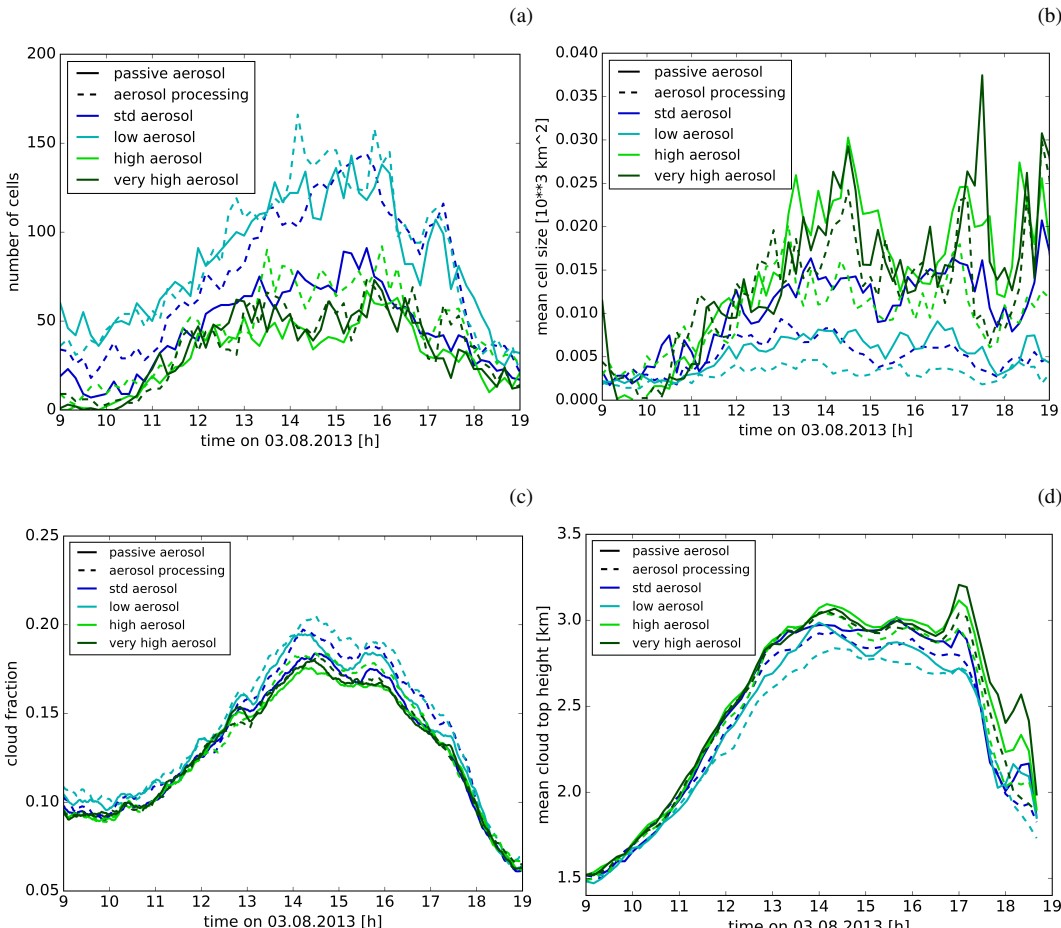

**Figure 5.** Time evolution of (a) number of cells, (b) average cell size, (c) cloud fraction and (d) domain average cloud top height. Cloudy areas are defined as having a water path larger than $1\,\mathrm{g\,m^{-2}}$ and cells as areas with a column maximum radar reflectivity (composite radar reflectivity) larger than $25\,\mathrm{dBZ}$. Different line colours indicate the different aerosol initial conditions, solid lines correspond to simulations with passive aerosols and dashed lines to simulations with aerosol processing.





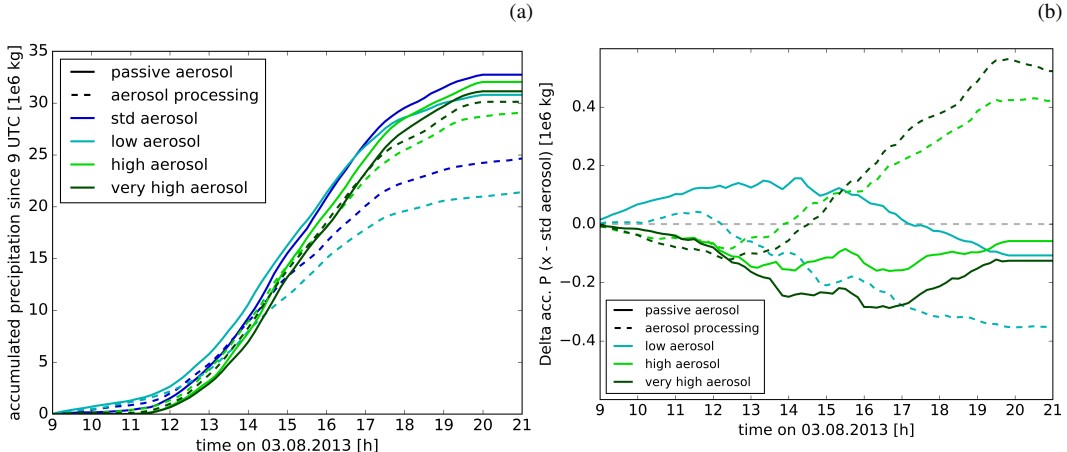

**Figure 6.** Time evolution of accumulated precipitation since 9 UTC (a) and its change in simulations with perturbed aerosol concentrations relative to the standard aerosol simulation (b). Different line colours indicate the different aerosol initial conditions, solid lines correspond to simulations with passive aerosols and dashed lines to simulations with aerosol processing.





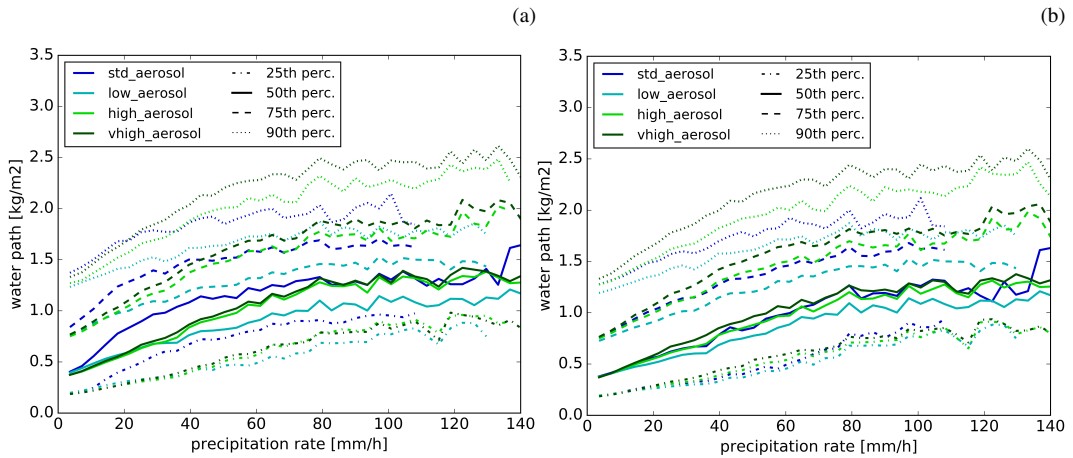

**Figure 7.** Percentiles of condensed water path (cloud and ice categories) in columns with certain precipitation rate for simulations with passive aerosol (a) and aerosol processing (b). Different colours correspond to the different aerosol profiles and different line styles to the different percentiles. Percentiles other than the median are only computed, if more than 100 data points are in a precipitation rate bin.



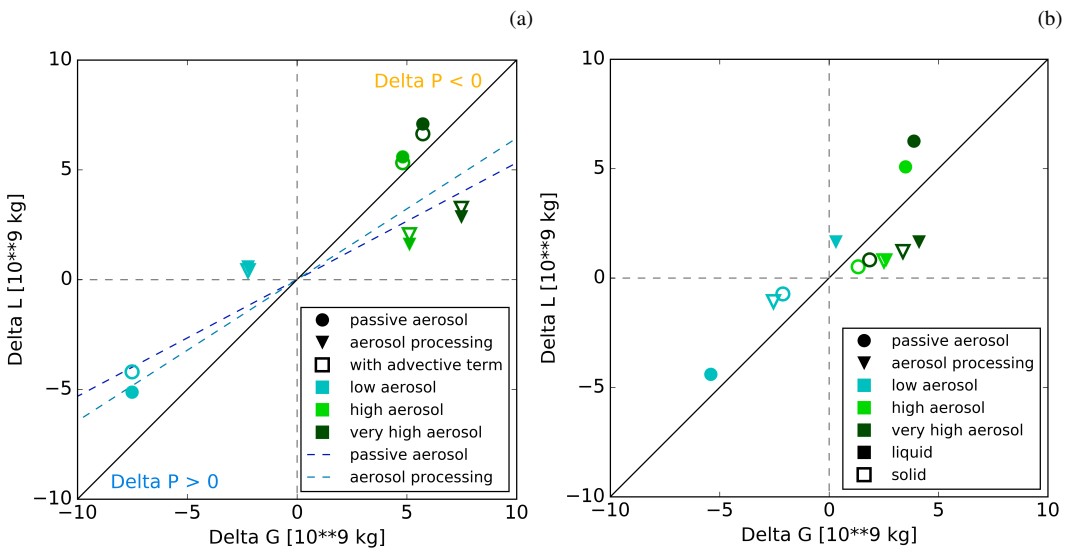

**Figure 8.** (a) Scatterplot of change in condensate gains and losses relative to the simulation with the standard aerosol profile. Points falling above the one-to-one line (solid black) portray a decrease in surface precipitation, while points below it portray a surface precipitation increase. The impact of advection of condensed water out of the domain is illustrated by the open symbols, for which the advective flux is discounted as loss. For points in the area between the solid black line and the dashed lines (dark blue: passive aerosol, light blue: aerosol processing) the change in condensate generation dominates over the change in precipitation efficiency (Appendix B). (b) Same as (a) but separating contribution of condensation and evaporation (filled symbols) from contribution of deposition and sublimation (open symbols).





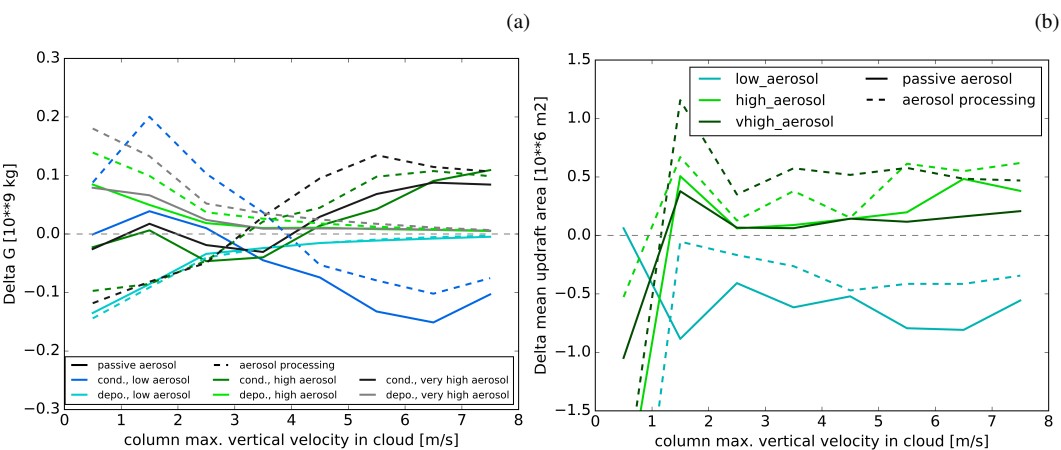

**Figure 9.** Characteristics of updraft regions classified by maximum column in-cloud vertical velocity: (a) Condensate gain and (b) updraft area. In (a) dark colours represent changes in condensation and lighter colours changes in deposition relative to the simulation with the control aerosol profile. Results from simulations with higher (lower) aerosol loading are depicted in greenish (blueish) colours. Solid lines correspond to simulations with passive aerosol, dashed lines to simulations with aerosol processing.





**Figure 10.** Average profiles of kinetic energy from vertical velocity (a,b), latent heat release (c,d) and condensate content (e,f). The left panels shows the average over all columns with a column maximum vertical velocity of $0-3\,\mathrm{m\,s^{-1}}$ and the right panels for those with a column maximum vertical velocity larger than $3\,\mathrm{m\,s^{-1}}$. The grey horizontal line indicates the location of the $0\,^{\circ}\mathrm{C}$ line in the different simulations.



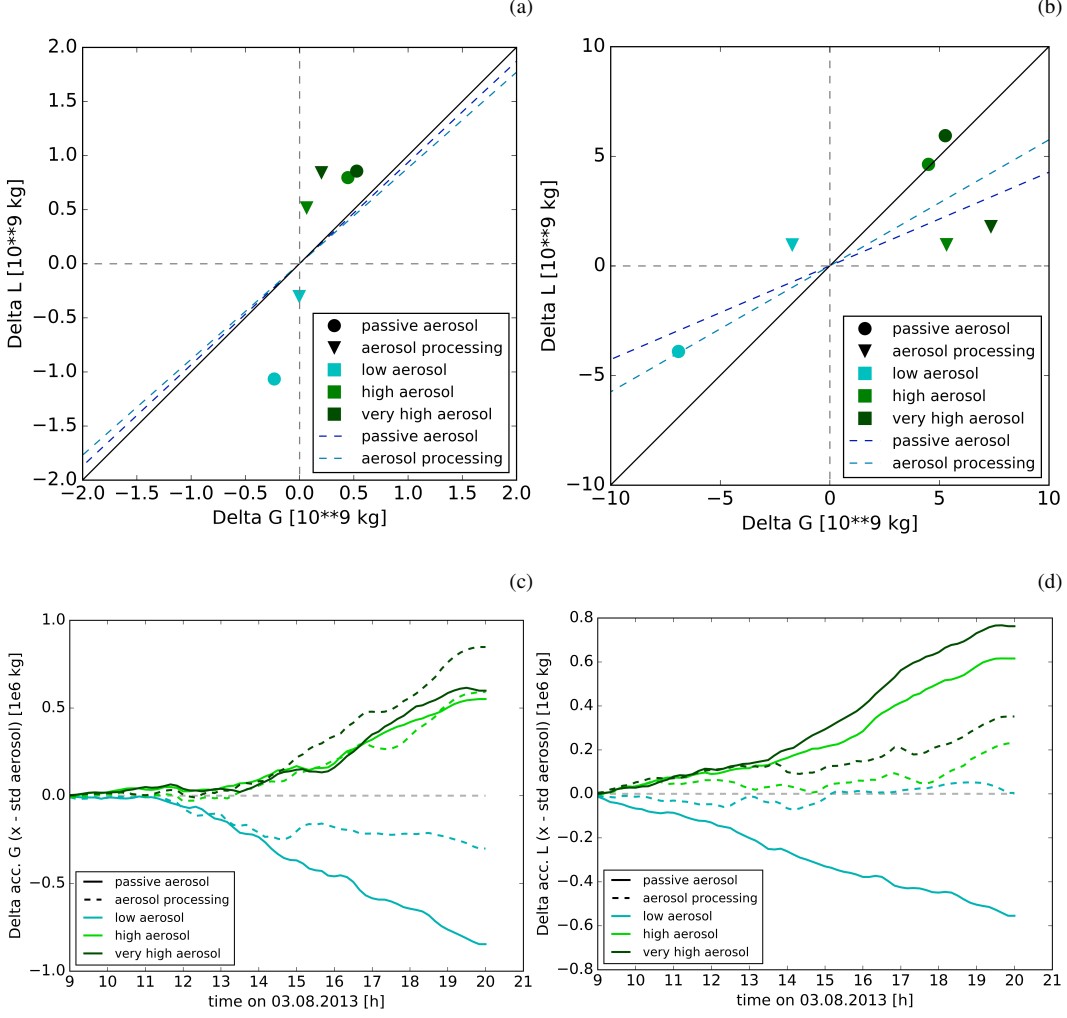

**Figure 11.** $\Delta$G in relation to $\Delta$L for the time period between 9-12 UTC (a) and 12-21 UTC (b). Difference in accumulated condensate generation (c) and condensate loss (d) relative to the simulation with standard aerosol profile. Solid lines represent simulations with passive aerosols and dashed lines those with aerosol processing.





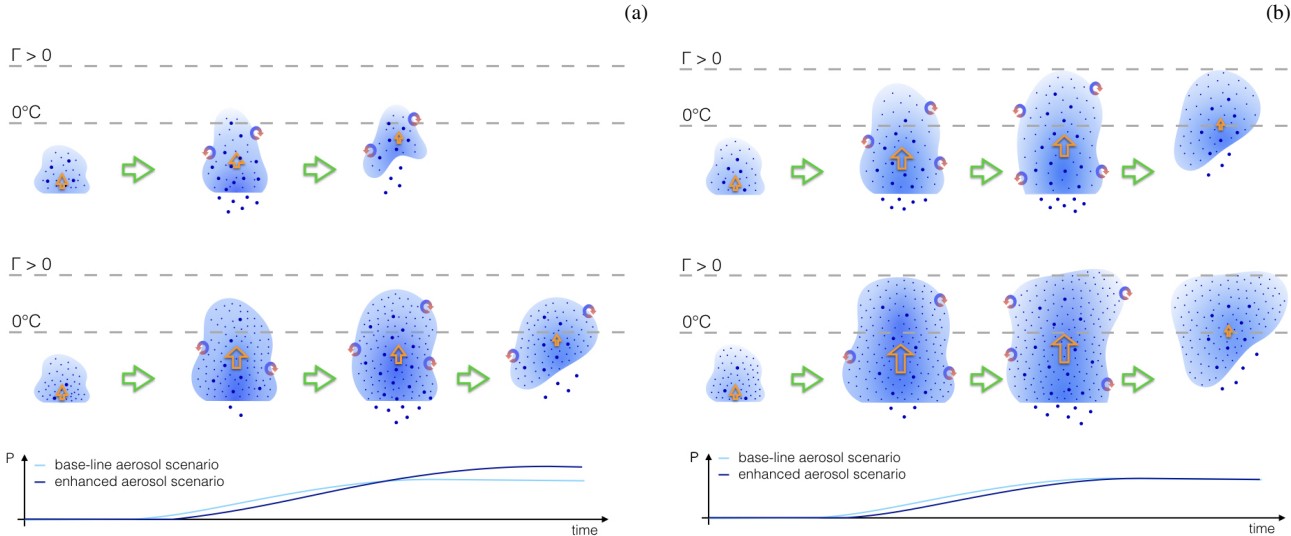

**Figure 12.** Schematic summary of aerosol-induced changes in the investigated clouds for a scenario, in which cloud tops are not limited by an upper level stable layer (left), and one, in which they are (right). The top row illustrates the cloud evolution in a low aerosol environment and the bottom one in a high aerosol environment. The intensity of the shading indicates the condensate mass mixing ratio. Small dots represent cloud droplets and larger ones rain drops. Different stages during of the cloud evolution are depicted from left to right and the lower section of each panel shows the time series of accumulated precipitation from each cloud (cyan: base-line aerosol case, dark blue: enhanced aerosol case). The orange arrows indicate vertical velocity in the convective core region.





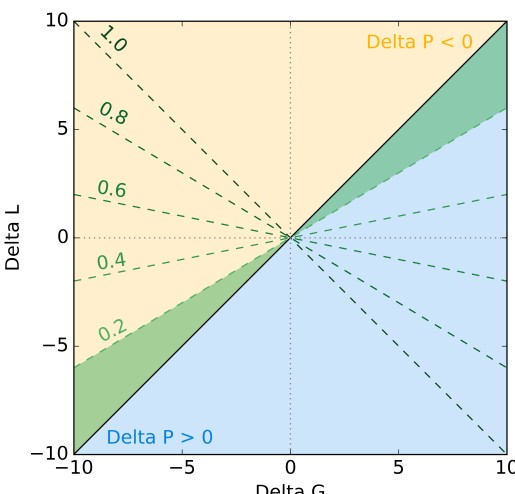

**Figure A1.** Exemplary plot of a ΔG versus ΔL diagram as explained in Appendix B. In simulations falling into the yellow (blue) shaded area less (more) precipitation is formed than in the reference. The change in precipitation is dominated by ΔG for simulations in the green shaded area, while precipitation changes in the rest of the phase space are dominated by changes in the precipitation efficiency. The green shaded area is for an example for a reference simulation with a precipitation efficiency of 0.2. The green dashed lines correspond to the indicated precipitation efficiency (0.2 to 1.9) of the reference simulation.





**Table 1.** Parameters used for the representation of the different hydrometeor types ($_x$: cloud, rain, ice, snow and graupel). The size distributions of all hydrometeors are described as gamma-distributions with a fixed curvature $\mu_x$. The relation between particle diameter $D_x$ and particle diameter $m_x$ is described by $m_x = c_x \cdot D_x^{d_x}$; the relation between $D_x$ and the terminal fall velocity by $v_x = a_x D_x^{b_x} \left( \frac{\rho_0}{\rho} \right)^{0.5}$.

|         | $\mu$ | $a_x$ [m$^{1-b_x}$s$^{-1}$] | $b_x$ [1] | $c_x$ [kg m$^{-3}$] | $d_x$ [1] |
|---------|-------|----------------------------|-----------|---------------------|-----------|
| cloud   | 0.0   | $3 \cdot 10^7$             | 2.0000    | $\frac{\pi}{6} \cdot 997$ | 3 |
| rain    | 2.5   | 130.00                     | 0.5000    | $\frac{\pi}{6} \cdot 997$ | 3 |
| ice     | 0.0   | 71.34                      | 0.6635    | $\frac{\pi}{6} \cdot 200$ | 3 |
| snow    | 2.5   | 4.84                       | 0.2500    | $\frac{\pi}{6} \cdot 100$ | 3 |
| graupel | 2.5   | 124.1                      | 0.6600    | $\frac{\pi}{6} \cdot 250$ | 3 |



**Table 2.** Parameters of the aerosol size distribution in the boundary layer prescribed in the initial and lateral boundary conditions.

|  | N [cm$^{-3}$] | m [kg m$^{-3}$] | $\sigma$ [1] |
|---|---|---|---|
| Aitken mode | 860 | $5.86 \cdot 10^{-10}$ | 2.2 |
| accumulation mode | 150 | $3.84 \cdot 10^{-9}$ | 1.7 |
| coarse mode | 0.23 | $1.07 \cdot 10^{-8}$ | 1.5 |
| insoluble aerosol | 16.7 | $4.26 \cdot 10^{-10}$ | 1.5 |



**Table 3.** Precipitation efficiency of the different simulations. Precipitation efficiency is defined as the ratio of domain integrated precipitation to domain integrated condensate gain (the sum of condensation and deposition rates. $1\,\mathrm{mg\,kg^{-1}}$.

|           | passive aerosol | aerosol processing |
|-----------|-----------------|--------------------|
| low       | 0.245           | 0.164              |
| std       | 0.246           | 0.185              |
| high      | 0.231           | 0.210              |
| very high | 0.224           | 0.215              |





**Table 4.** Change in surface precipitation expected from the simulated change in precipitation efficiency (left column) and the change in condensate generation (middle column). The last column gives the total relative change in precipitation, as predicted by the simulations.

|  |  | $G_{std}\Delta PE/P_{std}$ [%] | $PE_n\Delta G/P_{std}$ [%] | $\Delta P/P_{std}$ [%] |
|---|---|---|---|---|
| low | passive aerosol | 0.96 | -5.65 | -4.68 |
|  | aerosol processing | -11.13 | -1.50 | -12.63 |
| high | passive aerosol | -6.68 | 3.33 | -3.34 |
|  | aerosol processing | 12.39 | 4.35 | 16.74 |
| very high | passive aerosol | -9.61 | 3.85 | -5.76 |
|  | aerosol processing | 14.80 | 6.50 | 21.30 |