# Peer review of "Aerosol-cloud interactions in mixed-phase convective clouds. Part 1: Aerosol perturbations."

_Atmospheric Chemistry and Physics, 2017_

## Referee Comment (RC1) · Anonymous Referee #1 · 5 Oct 2017

Review of the manuscript 'Aerosol-cloud interactions in mixed-phase convective clouds. Part 1: Aerosol perturbations' by Miltenberger et al.

The authors use a newly-developed interactive cloud-aerosol microphysics scheme implemented in the Unified Model to perform high-resolution simulations of mixed-phase convective clouds which develop along a sea breeze convergence line. The simulations are evaluated against observations taken during the COPE field campaign. The simulations compare favourably to the observed thermodynamic and cloud properties. The impact of aerosol perturbations on simulated cloud properties, reflectivity and precipitation are investigated using the new microphysics scheme, as is the impact of passive aerosol vs allowing cloud-processing of aerosol. Including cloud processing of aerosol generally leads to increased model performance in terms of cloud base cloud droplet number concentration (CDNC) and reflectivity values but leads to an underestimation of domain-averaged surface precipitation rates and smaller convective cell sizes and numbers, although total cloud fraction is relatively unaffected. Although cloud processing of aerosol leads to changes in the cloud geometry, it does not affect the mechanisms of the cloud response to aerosol perturbations. Under an enhanced aerosol scenario, precipitation is initially suppressed but becomes enhanced once convection becomes organised. This is mostly due to modifications to condensate generation in the liquid-phase part of the cloud, and due to changes in precipitation efficiency. The authors report increased vertical velocities in convective cores and increased condensate production under enhanced aerosol environments ("convective invigoration"), but precipitation enhancement resulting from convective invigoration is limited by thermodynamic constraints on the cloud depth.

The manuscript is generally very well-written and uses a novel analysis technique of decomposing the precipitation response to changes in condensate generation and loss in order to identify and quantify the impact of aerosol perturbations. I find the results interesting and the study makes a good contribution to the body of literature on aerosol-cloud interactions.

However, the manuscript contains some confusing statements in places. There are also several claims made about aerosol-cloud microphysical processes occurring in the model and between the model configurations without figures or discussion of results provided to support these claims. Further, I found that the structure introducing the results of the perturbation experiments does not flow particularly well, and the first discussion of the perturbation results is somewhat obscured.

I have some concerns about the impact of a modelling framework that steps down directly from global model to a 1km grid, and would like to see that this doesn't have an impact on the high-resolution nests. Further, I would like to know whether the 250 m grid length used in this study is appropriate for the particular clouds investigated, as it has been shown that increasing spatial resolution does not necessarily lead to better representation of simulated storm morphology (e.g. Stein et al. 2014, 2015).

I have provided a list of general comments below. I recommend publication in ACP subject to major revisions.

**General comments:**

Introduction:
1. The paper focuses on simulations performed with a new bulk microphysics scheme with explicit aerosol processing. The literature cited in the introduction discusses simulated cloud response to aerosol but the studies cited include both bin and bulk schemes. It would therefore be good to note in the introduction which of these studies cited use bin schemes and which use bulk schemes.

Data and Methods:
2. The modelling framework used by the authors steps down from global (N512 resolution) to a 1 km nest, without stepping down through coarser outer nests. Can the authors show that this doesn't lead to any spurious artefacts either in the 1 km domain or in the boundary conditions for the 250 m generated from the 1 km nest? Especially with the 1 km nest containing land boundaries on its NW and SE sides, I have some concerns that stepping down from global to relatively fine resolution could have an undesired impact on the high-resolution domains.

3. The use of a 250 m grid length for the analysis domain: It has been shown that increasing spatial resolution does not necessarily lead to better representation of simulated storm morphology, particularly with respect to the width and intensity of simulated storm structures compared to those observed (e.g. Stein et al. 2014, 2015). Can the authors show that the choice of a 250 m grid length in the simulations presented in the paper is appropriate (compared to observations), compared to other grid lengths? Do the authors know whether the simulated storm structures in the current study have converged at the 250 m grid length used?

Results:
4. Model spinup and early isolated cells: Is it possible that the delayed precipitation development in the simulation compared to the observations, especially the generation of isolated cells early by the model which remain small and do not produce surface precipitation, is because the model is not fully spun up at this time? If not, is there another explanation for the lack of precipitation from these isolated cells compared to observations, given the relatively good agreement between the precipitation from the organised convection in the model and observations at later times?

5. Full distributions are presented and statements such as 'the underestimation of domain average precipitation is related to a smaller extent of weakly precipitating areas' are inferred (e.g. P9 L1: "is related to a smaller extent of weakly precipitating areas"). However, it is not possible to make conclusions on area / extent from the distributions alone as the full distributions contain both spatial and temporal components. That is, from the precipitation rate distribution alone it is not possible to distinguish whether the model underestimates precip rates compared to the radar observations (a) because there are fewer occurrences of

cloud in the model compared to the observed cloud, but which have the same precip rates as the observed cloud, or (b) whether there is the same amount of cloud in the model as that observed but with weaker precip rates compared to observations, or (c) a combination of less cloud with weaker precip rates. Are you able to show surface precip rates averaged below-cloud only, or similar figure comparisons, to distinguish between these potential cases? Otherwise, it may be more appropriate to phrase such statements in terms of e.g. "a reduced frequency of weakly precipitating points".

6. When the authors compare radar-derived and simulated rain rates, the claim is made several times in the manuscript that because the overall agreement between observed and modelled radar reflectivity distributions is better than that seen between the radar derived rain rate and modelled rain rate, this suggests potential problems with the radar derived surface precipitation rate for medium to low precipitation rates.

Whilst I agree that this is possible, could this not also be due to differences in the way that dBZ is calculated from the radar and from the model? i.e. could it not be that the radar-derived rain rates are correct (even if the model doesn't agree with them) and the simulated reflectivity values are wrong (even if they agree with the radar, i.e. the model appears to agree but for the wrong reasons)?

7. The authors make many statements on the processes responsible for certain behaviors. Examples are "these changes are due to the depletion of interstitial aerosols inside the cloud in the runs with aerosol processing, which impedes secondary activation in the model" (P11 L31 - 32) ; "While the Aitken mode is depleted downstream of convective cells, the accumulation mode increases due to evaporative release of aerosol. The collision-coalescence processes in the cloud lead to a transfer of aerosol from the Aitken to the accumulation mode. The coarse mode aerosol is increasing in cloudy areas mainly due to sub-cloud evaporation of rain and downstream of convective cells." (P11 L4-6), and similar instances occur throughout the text. However, no further information is given to justify these statements.

Are the authors able to provide comparisons of process rates or of e.g. interstitial aerosol amounts or of sub-cloud evaporation (in the first and second examples given above, respectively) to back up each of these statements? Even one table showing the difference in the aerosol process rates or amounts of (interstitial aerosol, secondary activation, etc) for the passive vs processed aerosol runs would make this immediately clear to the reader.

Further, the authors refer to depletion, enhancement and relative increases, but only show figures of fields from the simulation with aerosol processing. Are the authors referring to a description of the fields in this run only (e.g. depletion of aerosol in-cloud vs outside of cloud in the processed aerosol case), or a comparison with the passive aerosol run (e.g. in-cloud depletion of aerosol in the processed case vs the passive case)?

8. In general the discussion of the perturbed aerosol conditions does not flow particularly well. It is first introduced, with no figure reference, in section 4.1 which discusses aerosol processing vs passive aerosol treatment, and thus I found the perturbed aerosol discussion

somewhat obscured. Would it be possible to have a separate subsection discussing the impact of perturbed aerosol conditions, to make it clearer to the reader?

9. It would be useful to see SI Fig 7 c - f also shown as difference plots against the passive aerosol run to show the impact of cloud aerosol processing on the cloud and aerosol fields.

**Specific comments:**
P2 L1 - 'and modifications' should be 'modifications'
P2 L21-22 - it would be good to give some examples of studies which disagree on magnitude and/or sign.
P2 L29 - 'dominated with increasing environmental relative humidity' doesn't make sense.
P3 L10 - 'This conceptual idea has been developed using simulations of individual clouds under idealised conditions' - can you cite examples here?
P3 L13-15 'Also, Johnson et al. (2015) demonstrated that the precipitation signal is dependent on the values of uncertain parameters within the cloud microphysics parameterisation (parameter uncertainty).' - are these uncertain parameters inherently uncertain (variability in the parameters themselves), or uncertain because their values are unknown?
P4 L9 - 'provided' -> 'provides'
P4 L26 - 'Part 2' - presumably part 2 is a second paper to follow the present paper?
P5 L7 - 'Fig 1a' - I think the authors refer to Fig 1b.
P5 L22 - 'Fig 1b' - I think the authors refer to Fig. 1a.
P5 L23 - 'mean boundary layer top'? 'Top of the model' sounds like the authors refer to the model top.
P5 L27 - 'the residence time of air in the model domain is only several hours' - is this shown anywhere?
P5 L32 - 'aerosols' -> 'aerosol'
P6 L17-18 - 'The simulated precipitation rate and reflectivity distributions are particularly sensitive to the assumed graupel density and diameter-fall speed relation' - is this shown anywhere?
P6 L27 - 'in-soluble' -> 'insoluble'
P7 L20 - 'in the sections' -> 'in sections'
P8 L14 - SI Fig.s 1 and 2: it would perhaps be useful to the reader to have these in the main body of the paper rather than the supplementary information, for the reader to get a qualitative feel for the model behaviour in the first figure presented.
P8 L17-18 - 'While the majority of clouds develop along the convergence lines': for the simulation data it would be useful to have the convergence lines plotted on the Figures along with the reflectivity and coastline.
P8 L18-19 - 'A double-line feature also appears in model simulations': I find it hard to agree with this in some of the figures, especially the no aerosol processing figures.
P8 L19 - 'satellite data' - there has been no satellite data presented or discussed in the paper thus far.
P8 L20 - 'north-westerly' - do the authors mean north-easterly? (or SW to NE?)
P8 L23-24 - see general comment about spin-up
P8 L27-28 - 'The smaller domain average precipitation in the model is mainly due to the overall smaller cell sizes' - what happens if you compare precipitation from cells only (i.e. not

domain average), or domain average weighted by cell fraction. Do you get better agreement between the model and observations?

P8 L30-31 - 'The cessation of precipitation is linked to the dissolution of the convergence lines': Again, it would be useful to have the convergence lines plotted on Fig.s SI 1 and 2 to show this.

P9 L9 - 'The area covered' - see general comment 5

P9 L10 - 'The area covered by clouds with low reflectivity (< 10 dBZ) is underestimated in both simulations' - doesn't the processed aerosol case overestimate the occurrence of low reflectivity cloud?

P9 L16 - '...where the model underrepresents the medium surface precipitation rates' - refer to Fig. 3a here

P9 L16 - 18 - See general comment 6

P9 L22-24 - 'The maximum cloud depth in the observations only shows a small increase from about 5 km to about 5.5 − 6 km, while maximum cloud top height in the model increases from 3.5 km to 5 − 5.5 km' - see general point 4: could the delayed / weaker cloud development in the model be due to spinup?

P9 L28 - 'The larger maximum cloud top heights in the radar observations are mainly due to higher level ice clouds likely forming outside the model domain' - then why not restrict the analysis of the observations to the same region as the model domain analysed?

P10 L11 - Fig SI 5a - What is the high frequency occurrence of larger CDNC values high above the cloud base (5 to 7 km) which is seen in the passive aerosol case but not the processed case?

P10 L30 - 'will be robust' - robust compared to what? There are no observations to support the aerosol perturbation experiments, so any conclusions made about the processes that occur in the perturbation experiments can only be made in the context of the model with respect to itself.

P11 L4 - 'Aitken and accumulation mode aerosol concentrations are reduced inside the clouds due to CCN 5 activation.' - I can't see this easily from Fig 4. The accumulation mode seems to increase, not decrease.

P11 L6-8 - see general comment 7

P11 L6 - 'in the cloud' -> 'in the clouds'

P11 L16 - 'The Aitken mode is depleted within cloud' - is it actually depleted compared to the no processing case (which I imagine it is) or is there just a smaller aerosol number compared to other regions? Can you give a figure for each case (processed vs passive aerosol) or a difference figure to show the depletion?

P11 L22 - 'but the relative increases are more wide-spread and have a larger amplitude' - are these relative increases compared to another run? How are the relative increases shown?

P11 L24 - 29: Comparison of aerosol processing and aerosol concentrations. It is not clear which figure shows the passive vs processed aerosol statement in the first sentence. Also the rest of this paragraph discusses perturbations in aerosol concentration, but this has not yet been discussed and it is not clear which figure shows the aerosol perturbation results for the passive vs processed case discussed in this paragraph. (See general comment 8)

P11 L31-32 - 'These changes are due to the depletion of interstitial aerosols inside the cloud in the runs with processing, which impedes secondary activation in the model' - how do we know this? (See general comment 7).

P12 L3 - 'The impact of aerosol processing on aerosol fields and hydrometeor number concentrations is qualitatively very similar for simulations with perturbed aerosol initial conditions, except for the low aerosol simulations (not shown)' - the results of the aerosol perturbations haven't really been introduced (see general comment 8).

P12 L11 - how is a cell or cell size defined?

P12 L13-14 - 'Aerosol induced changes in cell number and size are smaller for aerosol concentrations enhanced above the standard aerosol profile compared to reduced aerosol concentrations' - does this indicate the transition from a CCN-limited to a dynamically-limited (updraft-limited) situation?

P12 L24-25 - 'For enhanced aerosol concentrations changes in cloud top height are larger in the aerosol processing than the passive aerosol simulation.' - this is interesting! Are you able to explain why?

P12 L26 - reference to SI Fig 8 - I think the authors mean to refer to Si Fig 6 here.

P13 L31 - 'seized' -> 'sized'

P14 L6 - 'is increasing' -> 'increasing'

P14 L14-15 - 'We hypothesise that the signal is consistent with parcel models in the highest percentile, because these correspond to updraft regions in clouds at the mature lifecycle stage, for which condensate production is compensating for losses due to precipitation production.' - could you help verify this by also showing condensed water path as a function of precip rate from the mature updrafts only?

P15 L10 - 'correspond to simulations, for which' -> 'correspond to simulations for which'

P16 L5 - 'importance For' -> 'importance. For'

P17 L1-2 - 'the vertical velocity is almost unaltered as is the cloud base temperature' - which figure shows this?

P17 L7-8 - 'The higher condensate amounts towards cloud top are also supported by slower conversion rates of cloud condensate into rain' - this is only true for the high vs standard aerosol case?

P17 L15-19 - How is this shown? (See general comment 7)

P18 L33-P19 L1 - see general comment 6

P19 L10 - remove commas from sentence in bullet point 2

P19 L12-13 - remove commas from first sentence in bullet point 3

P19 L16 - 'feedback mechanism, which' -> 'feedback mechanism which'

P19 L19-21 - this sentence is quite complicated and has too many commas. Can you simplify? Also 'two.way' -> 'two-way'

P20 L15 - 'hypothesis' -> 'hypothesise'

P21 Eqn A2 - per, ctr are not defined (I assume they mean 'perturbation' and 'control'?)

P22 L2 - remove commas in this sentence

**Figures:**

Fig 1:
- The image quality in the inset panel in 1a is not good, I find it hard to see in my printed copy.
- 'Aitken model' -> 'Aitken mode'
- 'bar showing' -> 'bar shows'

Fig 2:

- Fig 2a caption needs to state this is a timeseries

Fig 3:
- Panels (a) and (b0 are not labelled (I assume (a) is passive and (b) is processed aerosol case)

Fig 12:
- Can you label the columns 'uncapped' / 'capped' and the rows 'low / high'?
- Caption: 'scenario, in' -> 'scenario in'
- Caption: 'one, in' -> 'one in'
- Caption: 'during of the cloud' -> 'during the cloud'

Fig A1:
- 'Appendix B' -> 'Appendix A' ?

SI Fig 1, 2, 8:
- It is hard to see the land outline in my printed copy
- I would like to see convergence (even just a single contour) plotted on the model figures so that the reader can identify lines of convergence relative to the reflectivity.

SI Fig 3;
- The black lines (solid / dash for passive / processed aerosol) in the legend are confusing. I was expecting to see a black dashed and black solid line in the distribution in Fig 3a. I would remove these lines from the legend and just put the description in the figure caption.
- What is 'aerosol processing new' (labelled in the legend)?

SI Fig 4:
- What is the difference between sb and ml in Fig 5d?

SI Fig 7:
- 'redish' -> 'reddish'

SI Fig 8:
- I would find it easier to compare the panels in this figure if each panel had an extra caption describing the aerosol processing and concentration, e.g.: passive low, passive high, processed low, processed high

SI Fig 11:
- Caption: '(c) all hydrometeor' -> '(a) all hydrometeor'

SI Fig 12:
- 'Change mean' -> 'Change in mean'

SI Fig 15:
- (b), (d) - what happens in the high aerosol case between 6 and 7 km? There is a broken line.

- This figure isn't referred to in the manuscript

SI Fig 16:
- Caption: typo in 'hydrometeors'
- Caption: 'indicate es' -> 'indicates'

SI Fig.s 17, 18:
- These figures are not referred to in the manuscript

---

## Referee Comment (RC2) · Anonymous Referee #2 · 6 Nov 2017

**Review of "Aerosol-cloud interactions in mixed-phase convective clouds. Part 1: Aerosol perturbations" – Miltenberger et al. (2017)**

This paper presents an analysis of high-resolution simulations conducted over the southwest of the UK for convection observed during the COPE field campaign. This work has the potential to contribute to the growing body of literature on aerosol-cloud-precipitation interactions in the context of convective clouds. However, there are some serious issues that need to be addressed before moving forward with publication in ACP. Some of this issues include but are not limited to the paper length (and correspondingly, number of figures), the presentation quality, grammar, and lack of justification for claims in the text. More details on these major issues are provided below.

**Major Concerns**

1. **Grammar**: I found the text to be quite difficult (if not impossible) to follow in places due to the very large number of significant grammar errors. While I would typically provide a detailed list of such errors and corrects; the number of mistakes is too large for such details at this point in the review processes. Thus, I provide a list of items for the authors to review:

    a. **Oxford comma**: The Oxford comma is inconsistently used in the paper, making the intended meaning often difficult to determine. I suggest that the authors consider using it throughout to make it very clear that a list is being defined versus a sub-clause that further defines a term or concept.

    b. **Hyphens**: Hyphens are also used inconsistently thought the paper. For example, "cloud-top" and "cloud top" are used. There are also places where hyphens are needed, e.g., "upper-level stable layer" instead of "upper level stable layer". Please review the use of hyphens, especially in compound adjectives.

    c. **Subject-verb agreement**: There are numerous sentences in the manuscript in which the subject is singular and the verb is plural (or vice versa). For example, on Page 1, Line 25, the subject is "response", which is singular, and the verb is "suggest", which is plural. Moreover, the singular form of verbs is used when the term "data" is the subject; however, "data" is plural. Please review and make changes throughout the paper. Also note that "reflectivity" is singular.

    d. **Punctuation**: In particular, commas are used incorrectly throughout the paper (in addition to the Oxford comma discussed above). In many cases, it makes it very difficult to read the sentence and gain a coherent understanding of the intended meaning. In some cases, the lack of commas results in run-on sentences. There were several sentences in the text that I had to read several times before I was finally able to understand the authors' intention. For example, when using a phrase that introduces a sentence, a comma should follow, such as "According to their analysis, the balance between…", a comma should precede the reference on Page 3, Line 27, "After about 11 UTC, clouds organized…". These are just examples. Another set of examples in which commas are misused but create run-ons is as follows (just examples), "The CASIM module provides options for one- or two-way coupling between aerosol properties and cloud properties, and simulations are performed in both modes" and "Boundary layer processes, including surface fluxes of moisture and heat, are parameterized with the blended boundary layer scheme (Locket et al., 2015), and sub-grid scale turbulent processes are represented…".

e. **Incomplete sentences**: Please ensure that all sentences are complete (subject and verb). For example, the text on Page 7, Lines 29-31, form two incomplete sentences.

2. **Lack of supporting evidence and number of figures**: There are many places in the text, primarily in the discussion of the results where a conclusion is drawn without supporting evidence. My initial suggestion would be to at least plot the fields of interest to confirm that the conclusions are true; however, there are already too many figures in the paper (not to mention that it is hard to follow the analysis because the referenced figures switch back and for from those presented in the main text and those in the supplementary material). I suggest that the authors think very carefully about what figures are absolutely important to telling their story. If a figure is mentioned in passing, remove it in favor of a figure that shows that the conclusions are robust. For example, on Page 9, Line 22, it is noted that convection deepens with larger convergence forming along the sea breeze lines. Can you show this in the simulations? Not all figures need to be direct model-obs comparisons; the model can be used to justify your conclusions and fill in the gaps where the observations are lacking sufficient information.

Furthermore, some of the figures selected for the manuscript are difficult to read (partially due to the incomplete information given, e.g., units—see comment below regarding units in general—and even just a lack of axis titles). For example, Fig. 4 and the corresponding text on Page 11, first paragraph, are very difficult to follow. Perhaps another figure format would better convey the results? Moreover, conclusions are drawn regarding process rates but these values are not shown. These rates are predicted by the model. Did you look at the rates to confirm the conclusions?

Along these lines, I suggest that as the authors consolidate the figures, that the text be consolidated. The paper is long (my best guess is ~10,000-12,000 words), and this is just Part 1. My opinion is that less is more in some cases; you do not need to discuss every variable; instead, focus on the results that are most relevant to the story that you want to tell and the biggest conclusions. Otherwise, the important implications are muddled.

3. **References**: There are several places in the text where references should be included but are missing. For example, on Page 2, Lines 11-12, a reference or several references should be included for this "concept". In the discussion of aerosol regeneration, several references could be included but are omitted. Consider referencing Xue et al. (2010, J. Atmos. Sci.) and Mitra et al. (1992, J. Aerosol Sci.), just to name a few. Moreover, there are errors in the list of references that should be addressed (e.g., n/a for page numbers)

4. **Analysis**: There are several places in the text where the authors simply describe a figure but provide not reasoning for the differences depicted in such figures. For example, in Section 4.3, I just kept asking myself "why?" If details regarding why differences are observed are omitted, then I suggest shortening the discussion of the relevant topics and focusing on other aspects of the simulations.

Furthermore, regarding the analysis of G and L, it appears that this is only applicable for a closed system. Based on my understanding of the simulations, this is not the case because moisture could (and should) be advected through the inner domain's boundaries. Thus,

vapor may condense in the domain but be lost through the boundaries; it appears as though this is not accounted for; moreover, it is unclear how important this is in terms of the main results of the paper.

**Minor Concerns**

1. In general, please be consistent with the verb tense in the paper. Present and past tense are used throughout the discussion of the results, making it hard to determine if the authors intended for a sentence to be a general idea or specifically related to the case study.
2. In general, the units are kind of a mess in the paper. There are many places where spaces are not present, making it difficult to figure out what the units are supposed to be. Also, the units in figures are missing in places or change from figure to figure (e.g., degrees east versus degrees west longitude; the later is preferable for the study area so that negative coordinates are not needed). Consider using inverse units throughout the paper and in figures. Also, the use of "**" to represent an exponent is odd for a manuscript.
3. Please review the subscripts and superscripts in the figures. The variables are not consistent between the main text and the figures because of differences in the use of subscripts and superscripts.
4. The naming convention used for the runs changes from one figure to the next.
5. Page 1, Lines 15-18: The definition of invigoration is not in line with how it is commonly presented in the literature, i.e., related to enhanced lofting of liquid above the freezing level where subsequent freezing increases latent heating aloft and increases buoyancy. Please revise accordingly.
6. Page 1, Line 21: What are the thermodynamic constraints?
7. Page 5, Line 9: Why is the model top set to 40 km? Most modeling studies of even the deepest convection in the troposphere use model tops of 20-25 km. This seems as though a lot of computational cost is wasted simulating nearly the entire stratosphere.
8. Page 6, Line 19: The density selected for graupel is quite low, especially compared to what is commonly used in microphysics schemes. I believe some additional justification is needed.
9. Page 6, Line 28: Number density is not a conserved variable; please explain.
10. Page 7, Line 7: Is the Abdul-Razzal and Ghan (2000) activation parameterization particularly applicable to high-resolution simulations of convection?
11. Page 8, Line 19: Where are the satellite data?
12. Page 9, Line 17: What is meant by "sub-cloud evaporation in the radar diagnostic"? Do you mean that the simulated radar reflectivity is somehow accounting for the model-predicted evaporation rate?
13. Page 9, Line 20: Why did you choose 18 dBZ? Do you have a reference for such a choice? It is later stated that there is sensitivity (albeit small) to this choice; this should be expanded upon to convince the reader that the results are really robust.
14. Figure 7: Why not use a box-and-whisker plot (or something similar); the way the model output is presented makes it difficult to really understand the figure.

**Other Concerns**

1. Page 1, Line 8: Change "match to observed" to "correspondence with observed.
2. Page 1, Line 1: Change "effect" to "affect".
3. Page 1, Line 1: Remove "and" at the end of the line.
4. Page 1, Line 5: Remove "The" at the beginning of the sentence.

5. Page 2, Line 21: Change "processes involved" to "relevant processes".
6. Page 4, Line 11: Remove either "including" or "e.g.," because including both is redundant.
7. Page 4, Line 14: Define "COPE".
8. Page 4, Line 15: Add "the" before "UK".
9. Page 4, Line 24: This sentence does not make sense.
10. Page 4, Lines 29-30: This sentence needs to be reworded.
11. Page 5, Lines 15-16: Consider just saying that the operational microphysics was replaced and omit the "in addition to the standard model code"; this should be obvious to the reader.
12. Page 5, Line 25: Change "simulations, because:" to "simulations because".
13. Page 6, Line 15: I believe that these are the zeroth and third moments.
14. Page 6, Lines 14-15: This sentence is confusing (perhaps it is just the lack of an Oxford comma), but I am not completely sure. Also, the use of "relation" and "relations" is confusing. Is there a single relation for everything?
15. Page 6, Line 27: Insoluble is not hyphenated.
16. Page 7, Line 16: Change "traced" to "tracked".
17. Page 7, Line 28: Change to "The initial aerosol conditions".
18. Page 16, Lines 24-25: This sentence needs to be reword because it appears as though you are defining depths with units of m/s.

---

## Author Response (AR1)

**Replies to review RC1**
'Aerosol-cloud interactions in mixed-phase convective clouds. Part 1: Aerosol perturbations' by Miltenberger et al.

**General comments**
1. The paper focuses on simulations performed with a new bulk microphysics scheme with explicit aerosol processing. The literature cited in the introduction discusses simulated cloud response to aerosol but the studies cited include both bin and bulk schemes. It would therefore be good to note in the introduction which of these studies cited use bin schemes and which use bulk schemes.
*reply: We have changed the introduction accordingly and explicitly stated for each citation, whether a bulk or a bin scheme was used.*

2. The modelling framework used by the authors steps down from global (N512 resolution) to a 1 km nest, without stepping down through coarser outer nests. Can the authors show that this doesn't lead to any spurious artefacts either in the 1 km domain or in the boundary conditions for the 250 m generated from the 1 km nest? Especially with the 1 km nest containing land boundaries on its NW and SE sides, I have some concerns that stepping down from global to relatively fine resolution could have an undesired impact on the high-resolution domains.
*reply: We do not see any artificial features such as gravity waves originating from flow adjustments from the coarser resolution in the 1km domain. The current operational set-up of the Unified Model at the UK Met Office does use no intermediate nests for downscaling from global to kilometre-scale models (Clark et al., 2016).*
*Also previous published studies with the UM have stepped down from global to kilometre-scale resolution as well and confirm that the UM is able to handle this transition (e.g., Field et al., 2017; Grosvenor et al. 2017) and did provide reasonable results. The study by Field et al. (2017) shows that the mesoscale features do not change strongly for simulations with grid spacings of 16 km to 1 km (all nested directly in the global UM model). Furthermore the modelled cloud field structure are comparable in the Field et al. (2014) (using intermediated nests) and the Field et al. (2017) (no intermediate nests).*

3. The use of a 250 m grid length for the analysis domain: It has been shown that increasing spatial resolution does not necessarily lead to better representation of simulated storm morphology, particularly with respect to the width and intensity of simulated storm structures compared to those observed (e.g. Stein et al. 2014, 2015). Can the authors show that the choice of a 250 m grid length in the simulations presented in the paper is appropriate (compared to observations), compared to other grid lengths? Do the authors know whether the simulated storm structures in the current study have converged at the 250 m grid length used?
*reply: Simulations with 1 km and 500 m horizontal grid spacing have also been tested, but they compare less well to observations then the presented simulations with 250 m grid spacing. The comparison against observed surface rain rate and radar reflectivity is shown in Fig. 1. Fig. 2 shows map plots of column maximum reflectivity for simulations with different grid spacings and the observations at 14 UTC.*
*We have not tested simulations with even higher horizontal resolution as it becomes very expensive to cover the necessary area for the formation of the sea-breeze front at, e.g., 125 m grid-spacing. We therefore do not know whether the model convergences. Given the satisfactory performance of the simulations compared to observational data and the general elusiveness of convergence (Stein et al. 2014, Stein et al. 2015, Hanley et al. 2015), we think the 250 m grid spacing is an appropriate choice balancing the size of the domain with a satisfactory representation of the convective clouds.*
*change to paper: We have included the comparison of observations to the lower resolution simulations in the Supplementary information of the paper and added some text in section 2.1 (p. 5, l. 13/14).*

[Figure]

**Fig. 1.** *Distribution of column maximum reflectivity (left) and surface precipitation rate (right) from observational data (red line) and simulations with different grid spacings (cold colours).*

[Figure]

**Fig. 2.** *Column maximum radar reflectivity at 14 UTC from observations (top left) and model simulations with grid spacings of 250 m (top right), 500 m (bottom left) and 1 km (bottom right).*

4. Model spinup and early isolated cells: Is it possible that the delayed precipitation development in the simulation compared to the observations, especially the generation of isolated cells early by the model which remain small and do not produce surface precipitation, is because the model is not fully spun up at this time? If not, is there another explanation for the lack of precipitation from these isolated cells compared to observations, given the relatively good agreement between the precipitation from the organised convection in the model and observations at later times?

*reply: It may be a possibility some of the differences is due to spin-up. Although the model is already running for 9 hours at the time we start comparing the simulation to the observational data. This should be enough spin-up time for an NWP model. We think the model fails to organise the small convective cells forming in the early morning hours to larger clouds due to the absence of the forcing from the sea-breeze convergence. Larger cells form in the observations, which propagate for longer distances. It is known that NWP models have problems to generate larger cells in weak forcing situations (e.g., Stein et al., 2014; Hanley et al., 2015).*

*Fig. 2 shows the column maximum reflectivity at 10 UTC (top row) and precipitation Hovmöller plots (bottom row). The Hovmöller plots suggest that there are cells initiated in the model also before the convective line forms, but they do not grow to the same sizes as in the observations. Change to paper: We have added a sentence pertaining to the potential reason for the underestimation of the precipitation early in the simulation in sec. 3.1 (p. 9, l. 1-6).*

[Figure]

**Fig. 3.** *top: column maximum radar reflectivity from observations (left) and model simulations (right) at 10 UTC. bottom: Hovmöller plot of accumulated precipitation from radar observations (left) and the model simulation (right).*

5. Full distributions are presented and statements such as 'the underestimation of domain average precipitation is related to a smaller extent of weakly precipitating areas' are inferred (e.g. P9 L1: "is related to a smaller extent of weakly precipitating areas"). However, it is not possible to make conclusions on area / extent from the distributions alone as the full distributions contain both spatial and temporal components. That is, from the precipitation rate distribution alone it is not possible to distinguish whether the model underestimates precip rates compared to the radar observations (a) because there are fewer occurrences of cloud in the model compared to the observed cloud, but which have the same precip rates as the observed cloud, or (b) whether there is the same amount of cloud in the model as that observed but with weaker precip rates compared to observations, or (c) a combination of less cloud with weaker precip rates. Are you able to show surface precip rates averaged below-cloud only, or similar figure comparisons, to distinguish between these potential cases? Otherwise, it may be more appropriate to phrase such statements in terms of e.g. "a reduced frequency of weakly precipitating points".

*reply: Thank you for raising this important issues.The distributions shown in the paper are including points with precipitation only. Distributions including all data points are shown in Fig. 4 a. Qualitatively the same behaviour occurs if distributions are normalised with all points in the domain and time or only those where precipitation occurs. This is also the case if distributions for individual times are considered (Fig 4 b). Therefore the original conclusion holds that there are (a) fewer instances in space of weakly precipitating points in cloud and (b) fewer precipitating points over all.*

*change to paper: We explicitly state in figure captions and in the text that distributions over cloudy points only are shown. We also comment on the consistency in behaviour in time and a larger underestimation of weak precipitation rates, if all points are considered (p. 9, l. 13-16 & l. 23-28).*

[Figure]

**Fig. 4.** *(a) Distribution of precipitation rate including also none precipitating grid boxes. The observations and the model simulation contains the same number of points, so this is equivalent to scaling with the number of grid points. (b) Distribution of number of grid points in each precipitation bin for each 10 min interval between 10 UTC and 18 UTC separately (individual lines).*

6. When the authors compare radar-derived and simulated rain rates, the claim is made several times in the manuscript that because the overall agreement between observed and modelled radar reflectivity distributions is better than that seen between the radar derived rain rate and modelled rain rate, this suggests potential problems with the radar derived surface precipitation rate for medium to low precipitation rates.
Whilst I agree that this is possible, could this not also be due to differences in the way that dBZ is calculated from the radar and from the model? i.e. could it not be that the radar-derived rain rates are correct (even if the model doesn't agree with them) and the simulated reflectivity values are wrong (even if they agree with the radar, i.e. the model appears to agree but for the wrong reasons)?
*reply: It is certainly possible that there are problems with the modelled radar reflectivity.*
*change to paper: We added some discussion reflecting this aspect in section 3.1 (p. 10, l. 4-6).*

7. The authors make many statements on the processes responsible for certain behaviors. Examples are "these changes are due to the depletion of interstitial aerosols inside the cloud in the runs with aerosol processing, which impedes secondary activation in the model" (P11 L31 - 32) ; "While the Aitken mode is depleted downstream of convective cells, the accumulation mode increases due to evaporative release of aerosol. The collision-coalescence processes in the cloud lead to a transfer of aerosol from the Aitken to the accumulation mode. The coarse mode aerosol is increasing in cloudy areas mainly due to sub-cloud evaporation of rain and downstream of convective cells." (P11 L4-6), and similar instances occur throughout the text. However, no further information is given to justify these statements.
Are the authors able to provide comparisons of process rates or of e.g. interstitial aerosol amounts or of sub-cloud evaporation (in the first and second examples given above, respectively) to back up each of these statements? Even one table showing the difference in the aerosol process rates or amounts of (interstitial aerosol, secondary activation, etc) for the passive vs processed aerosol runs would make this immediately clear to the reader.
Further, the authors refer to depletion, enhancement and relative increases, but only show figures of fields from the simulation with aerosol processing. Are the authors referring to a description of the fields in this run only (e.g. depletion of aerosol in-cloud vs outside of cloud in the processed aerosol case), or a comparison with the passive aerosol run (e.g. in-cloud depletion of aerosol in the processed case vs the passive case)?

*reply: We take this comment to refer to section 4.1, i.e., the discussion of differences between the simulations with passive aerosol and aerosol processing. In the passive aerosol simulations the aerosol fields are essentially identical to the upstream boundary conditions, i.e., any changes in the aerosol field relative to the upstream values (left hand side of the plots) in the Hovmöller plots (Fig. 4 in the paper) or the cross-sections (SI Fig. 7) can be interpreted as a result of the aerosol processing. We provide the respective plots for the passive aerosol runs to confirm this in Fig. 4 of the paper and SI Fig. 10, 11. Because of this passive nature of the aerosol field, the processes named as a cause for certain features of the field (e.g., less secondary activation, reduced interstitial aerosol, enhanced coarse mode) are the only processes in the model that act on the aerosol fields with the right sign in the considered region (in-cloud, below cloud, outflow). We therefore think it is not necessary to add another table or figure to the paper given that there are already 10+ figures.*
*Change to paper: We also reformulated section 4.1 to better reflect the reasoning behind the statements (clarifying there is no change to the aerosol fields in the simulations with passive aerosol and processes added for the aerosol processing run).*

8. In general the discussion of the perturbed aerosol conditions does not flow particularly well. It is first introduced, with no figure reference, in section 4.1 which discusses aerosol processing vs passive aerosol treatment, and thus I found the perturbed aerosol discussion somewhat obscured. Would it be possible to have a separate subsection discussing the impact of perturbed aerosol conditions, to make it clearer to the reader?
*Change to paper: We changed the sections such that the discussion of aerosol processing and description of aerosol induced changes in the cloud field are now in a separate sections (new section 4 and 5, respectively).*

9. It would be useful to see SI Fig 7 c - f also shown as difference plots against the passive aerosol run to show the impact of cloud aerosol processing on the cloud and aerosol fields.
*see reply to point 7 of the review*

**Specific comments**
*Reply: Thank you for these comments and for spotting the errors in the formulation!*
P2 L1 - 'and modifications' should be ,modifications'
*Change to paper: as suggested*

P2 L21-22 - it would be good to give some examples of studies which disagree on magnitude and/ or sign.
*Change to paper: We have added references to the Tao et al. JGR 2007 study, which shows different responses in surface precipitation for the different clouds with the same modelling framework. We also reference the Khain et al. 2009 paper, which summarises different studies with varied precipitation responses.*

P2 L29 - 'dominated with increasing environmental relative humidity' doesn't make sense.
*Change to paper: reformulated*

P3 L10 - 'This conceptual idea has been developed using simulations of individual clouds under idealised conditions' - can you cite examples here?
*Change to paper: We included references to the following studies: Khain et al. JAS 2004, Khain et al. JAS 2005, and Rosenfeld et al. 2008.*

P3 L13-15 'Also, Johnson et al. (2015) demonstrated that the precipitation signal is dependent on the values of uncertain parameters within the cloud microphysics parameterisation (parameter uncertainty).' - are these uncertain parameters inherently uncertain (variability in the parameters themselves), or uncertain because their values are unknown?
*Reply: In the Johnson et al. (2015) parameters that are inherently uncertain, e.g., the graupel density (if not parameterised), and unknown parameters, e.g., immersion freezing coefficient, have*

*been perturbed. Several parameters, as for example the immersion freezing coefficients, are inherently uncertain and unknown.*
*Change to paper: We changed the text to clarify this (p. 3, l. 19-20).*

P4 L9 - 'provided' -> 'provides'
*Change to paper: as suggested*

P4 L26 - 'Part 2' - presumably part 2 is a second paper to follow the present paper?
*reply: Yes.*

P5 L7 - 'Fig 1a' - I think the authors refer to Fig 1b.
P5 L22 - 'Fig 1b' - I think the authors refer to Fig. 1a.
P5 L23 - 'mean boundary layer top'? 'Top of the model' sounds like the authors refer to the model top.
*Change to paper: as suggested*

P5 L27 - 'the residence time of air in the model domain is only several hours' - is this shown anywhere?
*Change to paper: This was not shown. Added how this estimate was obtained (p. 6, l. 6)*

P5 L32 - 'aerosols' -> ,aerosol'
*Change to paper: as suggested*

P6 L17-18 - 'The simulated precipitation rate and reflectivity distributions are particularly sensitive to the assumed graupel density and diameter-fall speed relation' - is this shown anywhere?
*Change to paper: Was not shown. We do not show the respective figure in the paper or the SI, since there are already a lot of figures, this sensitivity has been documented in previous papers, and is not significant for the scientific conclusions in the paper. The figure is shown in Fig. 6 in this reply.*

[Figure]

**Fig. 6.** *Sensitivity of surface precipitation rate distribution to various graupel mass-diameter fallspeed relations. Compare the cyan, blue, magenta and red lines. The red line is for the original CASIM formulations, the magenta with a reduced fall speed, the blue with the Seifert and Beheng (2006) relation, the dashed cyan line for the Locatelli and Hobbs (1974) high graupel density relation, and the solid cyan line for the Locatelli and Hobbs (1974) medium graupel density.*

P6 L27 - 'in-soluble' -> 'insoluble'
P7 L20 - 'in the sections' -> 'in sections'
*Change to paper: as suggested*

P8 L14 - SI Figs. 1 and 2: it would perhaps be useful to the reader to have these in the main body of the paper rather than the supplementary information, for the reader to get a qualitative feel for the model behaviour in the first figure presented.
*Change to paper: We have moved two of the subplots (12 UTC & 14 UTC) for the passive aerosol simulation in the main part of the paper. For brevity the rest of the panels remain in the Supplementary Information. We also moved the plots showing the domain configuration and the aerosol size distribution into the SI in order to not to inflate the number of figures in the main paper. These are mainly of interest to other modellers and therefore we think they do not necessarily have to be in the main paper. All information on the aerosol size distribution and domain settings is still contained in either Tab. 2 or the text.*

P8 L17-18 - 'While the majority of clouds develop along the convergence lines': for the simulation data it would be useful to have the convergence lines plotted on the Figures along with the reflectivity and coastline.
*Change to paper: We have included the contours showing the convergence at 250m above ground in all map plots (new Fig. 1, SI Fig. 2, 3 & 9). In addition we included a time series plot of convergence in the Supplementary information.*

P8 L18-19 - 'A double-line feature also appears in model simulations': I find it hard to agree with this in some of the figures, especially the no aerosol processing figures.
*Reply: The reviewer is certainly right that the double-line feature is less pronounced in the simulations than in the observational data and certainly least clear in the passive aerosol run.*
*Change to paper: We have modified the text accordingly (p. 8, l. 28/29).*

P8 L19 - 'satellite data' - there has been no satellite data presented or discussed in the paper thus far.
*Reply / change to paper: The satellite data we have is from a geostationary satellite and therefore the image quality over south-west England is not very good. We prefer not to show the images in the paper. The satellite images provide very little additional information relative to the radar data and we therefore removed the references to the satellite data from the paper.*

P8 L20 - 'north-westerly' - do the authors mean north-easterly? (or SW to NE?)
*Change to paper: as suggested*

P8 L23-24 - see general comment about spin-up
*Reply / change to paper: see general comment 4*

P8 L27-28 - 'The smaller domain average precipitation in the model is mainly due to the overall smaller cell sizes' - what happens if you compare precipitation from cells only (i.e. not domain average), or domain average weighted by cell fraction. Do you get better agreement between the model and observations?
*Reply: Mean precipitation rate from cells only is shown in Fig. 5. In contrast to the domain-average precipitation, it overestimates precipitation. This is consistent with the conclusion, that the cloudy area is underestimated in the model compared to observations. The number of cells in the observations is comparable or smaller than the one in the model simulations (Fig. 5 bottom left). The mean cell is larger in the observations than in the model (Fig. 5 bottom right). Accordingly we conclude that the smaller cell sizes are causing the underestimation of precipitation.*
*Change to paper: We have included these figures in the SI and referenced them in the text (p. 9, l. 1-6).*

P8 L30-31 - 'The cessation of precipitation is linked to the dissolution of the convergence lines': Again, it would be useful to have the convergence lines plotted on Fig.s SI 1 and 2 to show this.
*Change to paper: We have included the contours showing the convergence at 250m above ground in all map plots (new Fig. 1, SI Fig. 2, 3 & 9). In addition we included a time series plot of convergence in the Supplementary information.*

P9 L9 - 'The area covered' - see general comment 5

P9 L10 - 'The area covered by clouds with low reflectivity (< 10 dBZ) is underestimated in both simulations' - doesn't the processed aerosol case overestimate the occurrence of low reflectivity cloud?

*Reply: both overestimate for 10(5)-20dBZ, but underestimate below*

*Change to paper: The text in the paper has been modified along the lines of the reply to general*

[Figure]

*comment 5 (p. 9, l. 23-27).*

P9 L16 - '...where the model underrepresents the medium surface precipitation rates' - refer to Fig. 3a here

*Change to paper: as suggested*

P9 L16 - 18 - See general comment 6

*Reply / change to paper: see general comment 6*

P9 L22-24 - 'The maximum cloud depth in the observations only shows a small increase from about 5 km to about 5.5 – 6 km, while maximum cloud top height in the model increases from 3.5 km to 5 – 5.5 km' - see general point 4: could the delayed / weaker cloud development in the model be due to spinup?

*Reply / change to paper: see general comment 4*

P9 L28 - 'The larger maximum cloud top heights in the radar observations are mainly due to higher level ice clouds likely forming outside the model domain' - then why not restrict the analysis of the observations to the same region as the model domain analysed?

*Reply: The regions used for the analysis of the observations and the model are the same. However, cirrus clouds are present in the observational data. If they form outside the domain of the high-resolution simulation and are not present in the global model simulations, they will not be present in the analysis domain.*
*Change to paper: Reformulated sentence: „… due to higher level ice clouds, which are not present in the model simulations."*

**Fig. 5.** *Average precipitation rate for points with non-zero precipitation (top). Comparison of cell number (bottom left) and mean size (bottom right) from model simulations and radar observations. Cells are defined as continuous areas of composite radar reflectivity larger than 25 dBZ.*

P10 L11 - Fig SI 5a - What is the high frequency occurrence of larger CDNC values high above the cloud base (5 to 7 km) which is seen in the passive aerosol case but not the processed case?
*Reply: These are some high level clouds forming towards the end of the simulation (after 1730 UTC). Where they overlap with lower-level clouds they are included in the composite CDNC profile.*
*Change to paper: We modified the figure to include only contributions from the lowest cloud in a given profile.*

P10 L30 - 'will be robust' - robust compared to what? There are no observations to support the aerosol perturbation experiments, so any conclusions made about the processes that occur in the perturbation experiments can only be made in the context of the model with respect to itself.
*Reply: We reformulated the sentence (p. 11, l. 20).*

P11 L4 - 'Aitken and accumulation mode aerosol concentrations are reduced inside the clouds due to CCN 5 activation.' - I can't see this easily from Fig 4. The accumulation mode seems to increase, not decrease.
*Change to paper: We replaced Fig. 4 (Hovmöller plots) by SI Fig. 7a, d-f (cross-sections). Fig. 4 is now in the supplementary information. We think this will make the argumentation more easy to follow. All figure references have been changed accordingly.*

P11 L6-8 - see general comment 7
*Reply / change to paper: see general comment 7*

P11 L6 - 'in the cloud' -> 'in the clouds'
*Change to paper: as suggested*

P11 L16 - 'The Aitken mode is depleted within cloud' - is it actually depleted compared to the no processing case (which I imagine it is) or is there just a smaller aerosol number compared to other regions? Can you give a figure for each case (processed vs passive aerosol) or a difference figure to show the depletion?
P11 L22 - 'but the relative increases are more wide-spread and have a larger amplitude' - are these relative increases compared to another run? How are the relative increases shown?
P11 L24 - 29: Comparison of aerosol processing and aerosol concentrations. It is not clear which figure shows the passive vs processed aerosol statement in the first sentence. Also the rest of this paragraph discusses perturbations in aerosol concentration, but this has not yet been discussed and it is not clear which figure shows the aerosol perturbation results for the passive vs processed case discussed in this paragraph. (See general comment 8)
P11 L31-32 - 'These changes are due to the depletion of interstitial aerosols inside the cloud in the runs with processing, which impedes secondary activation in the model' - how do we know this? (See general comment 7).
*Reply: s. general comment 7.*
*Change to paper: We have reformulated section 4.1 (section 4 in the revised paper) to clarify the points raised.*

P12 L3 - 'The impact of aerosol processing on aerosol fields and hydrometeor number concentrations is qualitatively very similar for simulations with perturbed aerosol initial conditions,

except for the low aerosol simulations (not shown)' - the results of the aerosol perturbations haven't really been introduced (see general comment 8).
*Reply / change to paper: see general comment 8*

P12 L11 - how is a cell or cell size defined?
*Change to paper: We transferred this information from the figure caption to the main text (remains in the figure caption as well!).*

P12 L13-14 - 'Aerosol induced changes in cell number and size are smaller for aerosol concentrations enhanced above the standard aerosol profile compared to reduced aerosol concentrations' - does this indicate the transition from a CCN-limited to a dynamically-limited (updraft-limited) situation?
*Reply: Yes.*
*Change to paper: We have made this more clear in the paper (e.g. p.16, l. 9/10; p. 21, l. 23/24).*

P12 L24-25 - 'For enhanced aerosol concentrations changes in cloud top height are larger in the aerosol processing than the passive aerosol simulation.' - this is interesting! Are you able to explain why?
*Reply: The reason is the different position of the cloud top height to the base of the stable layer. For simulations with passive aerosols the cloud top height is larger for the low aerosol scenario compared to the simulations with aerosol processing. It is therefore closer to the use of the stable layer and accordingly the change in cloud top height is smaller for enhanced aerosol concentrations. This was already explained in the original paper on p. 12, l. 27-32.*
*Change to paper: We have partly reformulated the paragraph in question to make the argumentation clearer (p. 13, l. 24 to p. 14, l. 5).*

P12 L26 - reference to SI Fig 8 - I think the authors mean to refer to Si Fig 6 here.
P13 L31 - 'seized' -> 'sized'
P14 L6 - 'is increasing' -> ,increasing'
*Change to paper: as suggested*

P14 L14-15 - 'We hypothesise that the signal is consistent with parcel models in the highest percentile, because these correspond to updraft regions in clouds at the mature lifecycle stage, for which condensate production is compensating for losses due to precipitation production.' - could you help verify this by also showing condensed water path as a function of precip rate from the mature updrafts only?
*Reply: This section has been reformulated according to suggestions by reviewer #2 (new section 4).*

P15 L10 - 'correspond to simulations, for which' -> 'correspond to simulations for which'
P16 L5 - 'importance For' -> 'importance. For'
*Change to paper: as suggested*

P17 L1-2 - 'the vertical velocity is almost unaltered as is the cloud base temperature' - which figure shows this?
*Reply: This was not explicitly shown for the cloud base temperature, which we now indicate. The small difference in vertical velocity can be inferred from the small difference in the kinetic energy profile in Fig. 10 b. We have modified the text to make this more clear (p. 18, l. 8).*

P17 L7-8 - 'The higher condensate amounts towards cloud top are also supported by slower conversion rates of cloud condensate into rain' - this is only true for the high vs standard aerosol case?
*Reply: This is true for increasing aerosol concentrations from the low to high aerosol scenario. The impact is small for the transition from the high to very high aerosol scenario, as there is hardly any rain above the 0ºC line in the high aerosol scenario already.*
*Change to paper: We added an additional sentence to reflect this (p. 18, l. 16-19).*

P17 L15-19 - How is this shown? (See general comment 7)

*Reply: We cannot show this very easily as we do not have flux data or mixing terms available. The statements in this section are mainly speculations about the most likely mechanism explaining the observed profiles.*

*Change to paper: We rephrased the section to reflect the speculative nature of the statements* (**p. 18, l. 20-35**).

P18 L33-P19 L1 - see general comment 6

*Reply: s. general comment 6.*

*Change to paper: We modified the sentence according to the comment.*

P19 L10 - remove commas from sentence in bullet point 2
P19 L12-13 - remove commas from first sentence in bullet point 3
P19 L16 - 'feedback mechanism, which' -> 'feedback mechanism which'
P19 L19-21 - this sentence is quite complicated and has too many commas. Can you simplify?
Also 'two.way' -> ,two-way'
P20 L15 - 'hypothesis' -> ,hypothesise'
P21 Eqn A2 - per, ctr are not defined (I assume they mean 'perturbation' and 'control'?)
P22 L2 - remove commas in this sentence

*Change to paper: as suggested*

Figures:
Fig 1:
- The image quality in the inset panel in 1a is not good, I find it hard to see in my printed copy.
- 'Aitken model' -> 'Aitken mode'
- 'bar showing' -> 'bar shows'

*Change to paper: as suggested. The inset is now a separate figure, so it should be well readable*

Fig 2:
- Fig 2a caption needs to state this is a timeseries

*Change to paper: as suggested*

Fig 3:
- Panels (a) and (b) are not labelled (I assume (a) is passive and (b) is processed aerosol case)

*Change to paper: as suggested*

Fig 12:
- Can you label the columns 'uncapped' / 'capped' and the rows 'low / high'?
- Caption: 'scenario, in' -> 'scenario in'
- Caption: 'one, in' -> 'one in'
- Caption: 'during of the cloud' -> 'during the cloud'

*Change to paper: As suggested.*

Fig A1:
- 'Appendix B' -> 'Appendix A' ?

*Change to paper: As suggested.*

Fig 1, 2, 8:
- It is hard to see the land outline in my printed copy
- I would like to see convergence (even just a single contour) plotted on the model figures so that the reader can identify lines of convergence relative to the reflectivity.

*Change to paper: As suggested.*

SI Fig 3:

- The black lines (solid / dash for passive / processed aerosol) in the legend are confusing. I was expecting to see a black dashed and black solid line in the distribution in Fig 3a. I would remove these lines from the legend and just put the description in the figure caption.
- What is 'aerosol processing new' (labelled in the legend)?
*Change to paper:* As suggested.

SI Fig 4:
- What is the difference between sb and ml in Fig 5d?
*Change to paper:* Added explanation to figure caption.

SI Fig 7:
- 'redish' -> 'reddish'
*Change to paper:* as suggested

SI Fig 8:
- I would find it easier to compare the panels in this figure if each panel had an extra caption describing the aerosol processing and concentration, e.g.: passive low, passive high, processed low, processed high
*Change to paper:* as suggested

SI Fig 11:
- Caption: '(c) all hydrometeor' -> '(a) all hydrometeor'
*Change to paper:* as suggested

SI Fig 12:
- 'Change mean' -> 'Change in mean'
*Change to paper:* as suggested

SI Fig 15:
- (b), (d) - what happens in the high aerosol case between 6 and 7 km? There is a broken line.
*Reply:* There is some isolated high level clouds in this simulation between 6 and 7 km.
- This figure isn't referred to in the manuscript
*Change to paper:* Figure has been removed.

SI Fig 16:
- Caption: typo in 'hydrometeors'
- Caption: 'indicate es' -> 'indicates'
*Change to paper:* as suggested

SI Figs. 17, 18:
- These figures are not referred to in the manuscript
*Change to paper:* Figure has been removed.

   a. **Oxford comma**: The Oxford comma is inconsistently used in the paper, making the intended meaning often difficult to determine. I suggest that the authors consider using it throughout to make it very clear that a list is being defined versus a sub-clause that further defines a term or concept.

   b. **Hyphens**: Hyphens are also used inconsistently thought the paper. For example, "cloud-top" and "cloud top" are used. There are also places where hyphens are needed, e.g., "upper-level stable layer" instead of "upper level stable layer". Please review the use of hyphens, especially in compound adjectives.

   c. **Subject-verb agreement**: There are numerous sentences in the manuscript in which the subject is singular and the verb is plural (or vice versa). For example, on Page 1, Line 25, the subject is "response", which is singular, and the verb is "suggest", which is plural. Moreover, the singular form of verbs is used when the term "data" is the subject; however, "data" is plural. Please review and make changes throughout the paper. Also note that "reflectivity" is singular.

   d. **Punctuation**: In particular, commas are used incorrectly throughout the paper (in addition to the Oxford comma discussed above). In many cases, it makes it very difficult to read the sentence and gain a coherent understanding of the intended meaning. In some cases, the lack of commas results in run-on sentences. There were several sentences in the text that I had to read several times before I was finally able to understand the authors' intention. For example, when using a phrase that introduces a sentence, a comma should follow, such as "According to their analysis, the balance between...", a comma should precede the reference on Page 3, Line 27, "After about 11 UTC, clouds organized...". These are just examples. Another set of examples in which commas are misused but create run-ons is as follows (just examples), "The CASIM module provides options for one- or two-way coupling between aerosol properties and cloud properties, and simulations are performed in both modes" and "Boundary layer processes, including surface fluxes of moisture and heat, are parameterized with the blended boundary layer scheme (Locket et al., 2015), and sub-grid scale turbulent processes are represented...".

   e. **Incomplete sentences**: Please ensure that all sentences are complete (subject and verb). For example, the text on Page 7, Lines 29-31, form two incomplete sentences.

*Change to paper: Thank you for pointing out these inconsistencies in the manuscript. We have checked the manuscript very carefully for the raised issues.*

2. **Lack of supporting evidence and number of figures**: There are many places in the text, primarily in the discussion of the results where a conclusion is drawn without supporting evidence. My initial suggestion would be to at least plot the fields of interest to confirm that the conclusions are true; however, there are already too many figures in the paper (not to mention that it is hard to follow the analysis because the referenced figures switch back and for from those presented in the main text and those in the supplementary material). I suggest that the authors think very carefully about what figures are absolutely important to telling their story. If a figure is mentioned in passing, remove it in favor of a figure that shows that the conclusions are robust.

*Change to paper: We have checked the text for missing evidence. Where necessary we have added figures (mainly to the SI). For examples please see the more detailed comments below. In other instances, we have clarified whether the statement is a hypothesis or indicated that the figures is not shown, e.g., for sensitivity tests in the initial set-up of the model (the graupel fall speed relation).*

For example, on Page 9, Line 22, it is noted that convection deepens with larger convergence forming along the sea breeze lines. Can you show this in the simulations? Not all figures need to be direct model-obs comparisons; the model can be used to justify your conclusions and fill in the gaps where the observations are lacking sufficient information.

*Change to paper: We have included the contours showing the convergence at 250m above ground in all map plots (new Fig. 1, SI Fig. 2, 3 & 9). In addition we included a time series plot of convergence in the Supplementary information.*

Furthermore, some of the figures selected for the manuscript are difficult to read (partially due to the incomplete information given, e.g., units—see comment below regarding units in general—and even just a lack of axis titles). For example, Fig. 4 and the corresponding text on Page 11, first paragraph, are very difficult to follow. Perhaps another figure format would better convey the results?

*Reply: We have double checked all figures include axis titles, whoever we could identify no plots with missing axis titles except map plots, for which longitude and latitude were not labelled.*
*We also checked that the unit notation is consistent throughout the paper.*
*Regarding the Hovmöller plots for column aerosol loading (Fig. 4), we will swap these with the cross-sections of the aerosol field (SI Fig. 7) and adjust the text on page 11 to make the discussions clearer.*
*Change to paper: Thanks for pointing this out. Checked axis titles and units. We swapped Fig. 4 and SI Fig. 7, as the latter is probably more intuitive for the reader. We modified the text in section 4 accordingly.*

Moreover, conclusions are drawn regarding process rates but these values are not shown. These rates are predicted by the model. Did you look at the rates to confirm the conclusions?
*Reply: We do not have the rate output available. The rates can be inferred by looking at the difference between the aerosol fields from the passive aerosol and the aerosol processing simulation. The aerosol fields in the passive aerosol run are only affected by advection and show that the low values inside clouds are due to activation to CCN. Due to the low aerosol values there can be no activation. If there was no secondary activation the vertical decrease in CDNC that occurs in the aerosol processing runs would be also expected in the runs with passive aerosol. This is the only mechanism that is able to replenish CDNC in the model.*
 *For the increases in accumulation and coarse mode aerosol below cloud base, at cloud edges and in dissipating clouds there is also only one possible explanation given the model formulation: The only process in the model that can increase aerosol concentrations (except convergence) is evaporation of hydrometeors.*
*Change to paper: We included cross-sections of the aerosol fields and aerosol Hovmöller plots for the passive aerosol run in the SI. In addition we modified the text in section 4.1 to make clear why we are able to draw these conclusions.*

Along these lines, I suggest that as the authors consolidate the figures, that the text be consolidated. The paper is long (my best guess is ~10,000-12,000 words), and this is just Part 1. My opinion is that less is more in some cases; you do not need to discuss every variable; instead, focus on the results that are most relevant to the story that you want to tell and the biggest conclusions. Otherwise, the important implications are muddled.
*Change to paper: We have shortened the text, where this was possible without altering the scientific message, e.g., new section 5 (was section 4.2 ff previously).*

**3. References**: There are several places in the text where references should be included but are missing. For example, on Page 2, Lines 11-12, a reference or several references should be included for this "concept". In the discussion of aerosol regeneration, several references could be included but are omitted. Consider referencing Xue et al. (2010, J. Atmos. Sci.) and Mitra et al. (1992, J. Aerosol Sci.), just to name a few.
*Change to paper: Thank you for pointing out these references.We have included additional references at the suggested locations.*

Moreover, there are errors in the list of references that should be addressed (e.g., n/a for page numbers)
*Change to paper: Thank you for spotting these errors. We checked the reference list and added the missing information.*

**4. Analysis**: There are several places in the text where the authors simply describe a figure but provide not reasoning for the differences depicted in such figures. For example, in Section 4.3, I just kept asking myself "why?" If details regarding why differences are observed are omitted, then I suggest shortening the discussion of the relevant topics and focusing on other aspects of the simulations.
*Change to paper: Section 4.3 was reformulated. We have checked the paper for instances of observations described without an explanation and made sure an explanation or hypothesis is provided whenever possible. It has been clarified that the previous section 4.2ff (now section 5) merely describes the changes in cloud field properties due to aerosol, while the physical mechanisms are discussed in the following section. We think that makes the argumentation easier to follow, as many changes are interlinked. Section 5 was shortened relative to the previous sections.*

Furthermore, regarding the analysis of G and L, it appears that this is only applicable for a closed system. Based on my understanding of the simulations, this is not the case because moisture could (and should) be advected through the inner domain's boundaries. Thus,
vapor may condense in the domain but be lost through the boundaries; it appears as though this is not accounted for; moreover, it is unclear how important this is in terms of the main results of the paper.
*Reply: The analysis of G and L is strictly only valid if there is no advection out of the domain, as the reviewer points out correctly. This is of course not the case in the presented model simulations. The effects this has are discussed in the original paper (paragraph starting at p. 14, line 30 in the original paper and Fig. 8a). It was found that there are only small differences in the advective terms between runs with different aerosol concentrations and we therefore think the analysis is valid even without considering the advective terms explicitly in the rest of the paper.*
*Change to paper: We reformulated parts of this paragraph in the paper to make the argumentation clearer (p. 15, l. 24 to p. 16, l. 1).*

**Minor Concerns**
1. In general, please be consistent with the verb tense in the paper. Present and past tense are used throughout the discussion of the results, making it hard to determine if the authors intended for a sentence to be a general idea or specifically related to the case study.
*Change to paper: We carefully checked the verb tense in the paper.*

2. In general, the units are kind of a mess in the paper. There are many places where spaces are not present, making it difficult to figure out what the units are supposed to be. Also, the units in figures are missing in places or change from figure to figure (e.g., degrees east versus degrees west longitude; the later is preferable for the study area so that negative coordinates are not needed). Consider using inverse units throughout the paper and in figures. Also, the use of "**" to represent an exponent is odd for a manuscript.
*Change to paper: We have check all units in the paper and ensured the typesetting is consistent throughout.*

3. Please review the subscripts and superscripts in the figures. The variables are not consistent between the main text and the figures because of differences in the use of subscripts and superscripts.
*Change to paper: We made all sub- and superscripts consistent between text and figures.*

4. The naming convention used for the runs changes from one figure to the next.
*Change to paper: We checked all the run names and ensured they are consistent.*

5. Page 1, Lines 15-18: The definition of invigoration is not in line with how it is commonly presented in the literature, i.e., related to enhanced lofting of liquid above the freezing level where subsequent freezing increases latent heating aloft and increases buoyancy. Please revise accordingly.

*Reply: We use the term „convective invigoration" for the occurrence of higher updrafts in convective clouds in high aerosol environments, which is consistent with the use of invigoration in the literature. However, in contrast to the common presentation in literature, we do not find strongly enhanced latent heating due to freezing. Instead latent heating in the warm-phase section of the clouds is enhanced. This partly contradicts the results presented so far in literature, but is consistent with the results of our modelling study and is supported by multiple figures.*

*Change to paper: We reformulated the sentence to better reflect, where our findings disagree with literature.*

6. Page 1, Line 21: What are the thermodynamic constraints?

*Change to paper: Replaced „thermodynamic constraints" with „stable layer in the upper troposphere".*

7. Page 5, Line 9: Why is the model top set to 40 km? Most modeling studies of even the deepest convection in the troposphere use model tops of 20-25 km. This seems as though a lot of computational cost is wasted simulating nearly the entire stratosphere.

*Reply: The model top is identical to that used in the operational configuration of the Unified Model for regional model simulations (UKV-nest with a grid-spacing of 1.5 km). Since changing the model top can impact the reflection of gravity waves from the upper boundary, we think it is best to use the set-up developed and tested in operational use.*

8. Page 6, Line 19: The density selected for graupel is quite low, especially compared to what is commonly used in microphysics schemes. I believe some additional justification is needed.

*Reply: We have tested a range of different graupel density and consistent diameter-fallspeed relations. The chosen relation provides the best agreement with the observed rain rate distribution. A figure showing the changes in the precipitation rate distribution for a number of graupel diameter-fallspeed relations is shown in Fig. 6 in reply to reviewer #1.*

*Change to paper: The paper does already state our main motivation for choosing this graupel density and corresponding diameter-fallspeed relation.*

9. Page 6, Line 28: Number density is not a conserved variable; please explain.

*Reply: We are not sure what this comment is referring to. It is not stated in the paper that number density is a conserved variable. The referenced line states that aerosol number density is a prognostic variable in UM-CASIM.*

10. Page 7, Line 7: Is the Abdul-Razzal and Ghan (2000) activation parameterization particularly applicable to high-resolution simulations of convection?

*Reply: The Abdul-Razzak and Ghan (2000) activation parameterisation is one of the published parameterisations for activation of cloud droplets from the aerosol based on the prognostic vertical velocity. It is therefore suitable for the purpose in the presented study. This parameterisation has been use in previous high-resolution model studies (e.g., Grosvenor et al., 2017).*

11. Page 8, Line 19: Where are the satellite data?

*Reply / change to paper: The satellite data we have is from a geostationary satellite and therefore the image quality over south-west England is not very good. We prefer not to show the images in the paper. The satellite images provide very little additional information relative to the radar data and we therefore removed the references to the satellite data from the paper.*

12. Page 9, Line 17: What is meant by "sub-cloud evaporation in the radar diagnostic"? Do you mean that the simulated radar reflectivity is somehow accounting for the model-predicted evaporation rate?

*Reply: Surface precipitation rate is derived from radar data by a fixed relationship between reflectivity and surface rain rate (a so-called Z-R formula; Harrison et al., 2016). However, sub-cloud evaporation can influence the surface precipitation rate and this is not taken into account very well by using a fixed Z-R formula (e.g., Li and Srivastava, 2001). In contrast, sub-cloud evaporation is explicit modelled in the NWP simulations. This difference in the representation of sub-cloud evaporation (or lack thereof) likely contributes to the different performance of the model simulations in the evaluation against radar-derived surface precipitation rates and radar reflectivity.
Change to paper: We reformulated the sentence for clarification (p. 9, l. 17/18).*

13. Page 9, Line 20: Why did you choose 18 dBZ? Do you have a reference for such a choice? It is later stated that there is sensitivity (albeit small) to this choice; this should be expanded upon to convince the reader that the results are really robust.
*Reply: 18 dBZ is often used in radar networks as threshold for the radar echo top (e.g., Lakshmanan et al., 2013; Scovell and Al-Sakka, 2016).
Change to paper: We included a quantitative description of the sensitivity to the threshold in the paper (p. 10, l. 17-19).*

14. Figure 7: Why not use a box-and-whisker plot (or something similar); the way the model output is presented makes it difficult to really understand the figure.
*Change to paper: Thank you for this suggestion. The new plots are indeed easier to read. We have changed the plots as suggested. Note that a bug in the code used to generate the original figures was detected after the submission of the paper. The updated plot has been included in the paper. The text in section 5.3 has been modified accordingly.*

**Other Concerns**
1. Page 1, Line 8: Change "match to observed" to "correspondence with observed".
2. Page 1, Line 1: Change "effect" to "affect".
3. Page 2, Line 1: Remove "and" at the end of the line.
4. Page 2, Line 5: Remove "The" at the beginning of the sentence.
5. Page 2, Line 21: Change "processes involved" to "relevant processes".
6. Page 4, Line 11: Remove either "including" or "e.g.," because including both is redundant.
7. Page 4, Line 14: Define "COPE".
8. Page 4, Line 15: Add "the" before "UK".
9. Page 4, Line 24: This sentence does not make sense.
10. Page 4, Lines 29-30: This sentence needs to be reworded.
11. Page 5, Lines 15-16: Consider just saying that the operational microphysics was replaced and omit the "in addition to the standard model code"; this should be obvious to the reader.
12. Page 5, Line 25: Change "simulations, because:" to "simulations because".
13. Page 6, Line 15: I believe that these are the zeroth and third moments.
14. Page 6, Lines 14-15: This sentence is confusing (perhaps it is just the lack of an Oxford comma), but I am not completely sure. Also, the use of "relation" and "relations" is confusing. Is there a single relation for everything?
15. Page 6, Line 27: Insoluble is not hyphenated.
16. Page 7, Line 16: Change "traced" to "tracked".
17. Page 7, Line 28: Change to "The initial aerosol conditions".
18. Page 16, Lines 24-25: This sentence needs to be reword because it appears as though you are defining depths with units of m/s.
*Change to paper: Thanks for spotting these. All suggested changes have been included and sentences have been reformulated.*

*Correspondence to:* Annette K. Miltenberger (a.miltenberger@leeds.ac.uk)

**Abstract.** Changes induced by perturbed aerosol conditions in moderately deep mixed-phase convective clouds (cloud top heigh $\sim 5$ km) developing along sea-breeze convergence lines are investigated with high resolution numerical model simulations. The simulations utilise the newly developed Cloud-AeroSol Interacting Microphysics module (CASIM) for the Unified Model, which allows for the representation of the two-way interaction between cloud and aerosol fields. Simulations are evaluated against observations collected during the Convective Precipitation Experiment (COPE) field campaign over the southwestern peninsula of the UK in 2013. The simulations compare favourably with observed thermodynamic profiles, cloud base cloud droplet number concentrations (CDNC), cloud depth, and radar reflectivity statistics. Including the modification of aerosol fields by cloud microphysical processes improves the correspondence with observed CDNC values and spatial variability, but reduces the agreement with observations for average cloud size and cloud top height.

Accumulated precipitation is suppressed for higher aerosol conditions before clouds become organised along the sea-breeze convergence lines. Changes in precipitation are smaller in simulations with aerosol processing. The precipitation suppression is due to less efficient precipitation production by warm-phase microphysics, consistent with parcel model predictions. In contrast, after convective cells organise along the sea-breeze convergence zone, accumulated precipitation increases with aerosol concentrations. Condensate production increases with the aerosol concentrations due to higher vertical velocities in the convective cores and higher cloud top heights. However, for the highest aerosol scenarios no further increase in the condensate production occurs, as clouds grow into an upper level stable layer. In these cases, the reduced precipitation efficiency dominates the precipitation response and no further precipitation enhancement occurs. Previous studies of deep convective clouds have related larger vertical velocities under high aerosol conditions to enhanced latent heating from freezing. In the presented simulations changes in latent heating above the $0\,°\text{C}$ are negligible, but latent heating from condensation increases with aerosol concentrations. It is hypothesised that this increase is related to changes in the cloud field structure reducing the mixing of environmental air into the convective core.

The precipitation response of the deeper mixed-phase clouds along well-established convergence lines can be the opposite of predictions from parcel models. This occurs when clouds interact with a pre-existing thermodynamic environment and cloud field structural changes occur that are not captured by simple parcel model approaches.

*Copyright statement.* The works published in this journal are distributed under the Creative Commons Attribution 3.0 License. This licence does not affect the Crown copyright work, which is reusable under the Open Government Licence (OGL). The Creative Commons Attribution 3.0 License and the OGL are interoperable and do not conflict with, reduce, or limit each other.

[revised manuscript text omitted]

**Figure 5.** (a) Comparison of the distribution of average precipitation rate for grid points with non-zero precipitation. (b) Timeseries of domain integrated convergence at 250 m above ground for the simulations with the standard aerosol scenario. (c) Cell number and (d) mean cell size from radar observations (red) and model simulations with passive aerosols (solid line) and aerosol processing (dashed line), respectively. Simulated precipitation rates have been coarse-grained to the spatial resolution of the radar observations (1 km horizontal). Cells are defined as continuous regions with column maximum reflectivity larger than 25 dBZ.

[Figure]

**Figure 6.** Distribution of surface precipitation rate for (a) gridpoints with surface precipitation and (b) all grid points. (c) Distribution of radar reflectivity at 750 m computed over grid points with reflectivity larger than 0 dBZ. Radar data is shown in red and model data in blue. (d) Distribution of column maximum radar reflectivty computed over all grid points. The solid line shows results from the simulation with passive aerosols and the dashed line from the simulation with aerosol processing. Simulated precipitation rates and radar reflectivity have been coarse-grained to the spatial resolution of the radar observations (1 km horizontal and 500 m vertical).

[Figure]

**Figure 7.** Comparison of (a) domain mean and (b) maximum height of the 18 dBZ contour from model simulations and radar data.

[Figure]

**Figure 8.** Cloud droplet number concentration from simulations with (a) passive aerosol and (b) aerosol processing as a function of altitude above cloud base. Cloud base is defined as the lowest model level in each column with a cloud droplet mass larger than $1\,\mathrm{mg\,kg^{-1}}$.

[Figure]

**Figure 9.** Thermodynamic profilesfrom (a) radiosonde released at Davidstow at 1200 UTC and (b) the closest grid point in the model for the simulation with the standard aerosol profile and passive aerosols.

[Figure]

**Figure 10.** Timeseries of (a) the 0 °C level height and (b) the lifting condensation level from the radiosondings at Davidstow (red diamonds) and the closest model gridpoint (blue lines). The lifting condensation level is computed based on air parcel having (i) the thermodynamical properties of the first model level above the surface (upper plot panel b) or (ii) the mean thermodynamical properties of the lowest 50 hPa (lower plot in panel b).

[Figure]

**Figure 11.** Aerosol fields from the simulation with aerosol processing at 14 UTC. (a) Colour shading shows the column maximum reflectivity and the black line indicates the location of the cross-sections plotted in the other panels: (b) number density of Aitken mode aerosol, (c) accumulation mode aerosol, and (d) coarse mode aerosol. The white contour lines in panels (b) to (d) indicate areas with hydrometeor mixing ratios larger than $1\,\mathrm{mg\,kg^{-1}}$.

[Figure]

**Figure 12.** Hovmöller diagrams of latitudinally averaged column integrated aerosol number density from the simulation with passive aerosol (left column) and aerosol processing (right column) and the standard aerosol profile: (a, b) Aitken , (c, d) accumulation, and (e, f) coarse mode. The white contour indicates areas where the condensed water path is larger than $0.1 \, \mathrm{kg \, m^{-2}}$.

[Figure]

**Figure 13.** Column maximum radar reflectivity over the COPE domain at 14 UTC from the simulations with (a, b) passive aerosol and (c, d) aerosol processing. The left panels are from simulations with the low aerosol number concentrations and the right panels from those with high aerosol number concentrations.

[Figure]

**Figure 14.** Distribution of (a) cloud top and (b) cloud base height in the different simulations. Cloud top (base) is defined as the highest (lowest) point in each column where the sum of cloud and ice water content exceeds $1\,\mathrm{mg\,kg^{-1}}$.

[Figure]

**Figure 15.** The precipitation rate distribution between 9 UTC and 21 UTC. Different line colours indicate the different aerosol initial conditions, solid lines correspond to simulations with passive aerosols and dashed lines to simulations with aerosol processing.

[Figure]

**Figure 16.** Time evolution of condensed water path in (a) all hydrometeor categories, (b) the cloud, ice and snow categories, and (c) the graupel and rain categories. Different line colours indicate the different aerosol initial conditions, solid lines correspond to simulations with passive aerosols and dashed lines to simulations with aerosol processing.

[Figure]

**Figure 17.** Percentiles of condensed water path (including cloud droplet and ice categories) for the time period between (a, b) 9 - 12 UTC and (c, d) 12 - 20 UTC in columns with certain precipitation rate. (e, f) Percentiles of condensed water path (all hydrometeor categories) in columns with certain precipitation rate for the entire simulation period. The left column shows the simulations with passive aerosol and the right column those with aerosol processing. Different colours correspond to the different aerosol profiles. Values are only shown for precipitation bins with more than 100 data points. The boxes cover the interquartile range ($25^{th}$ to $75^{th}$ percentile). The whiskers represent the $10^{th}$ and $90^{th}$ percentile, respectively. The median water path is shown by the horizontal line in the box.

[Figure]

**Figure 18.** (a) Mean depth of updraft regions ($w > 0\,\mathrm{m\,s^{-1}}$ and condensate content larger $1\,\mathrm{mg\,kg^{-1}}$) classified by maximum column in-cloud vertical velocity. (b) Change in mean condensation, evaporation, deposition and sublimation rates for each maximum in-cloud vertical velocity bin. Results from simulations with higher (lower) aerosol loading are depicted in green (blue). Solid lines correspond to simulations with passive aerosol, dashed lines to simulations with aerosol processing.

[Figure]

**Figure 19.** Average profiles of mean mixing ratio of the different hydrometeors (colours) for simulations with (a, b) low, (c, d) standard, and (e, f) high aerosol concentrations. The left panels shows the average over all columns with a column maximum vertical velocity of $0-3\,\mathrm{m\,s^{-1}}$ and the right panels for those with a column maximum vertical velocity larger than $3\,\mathrm{m\,s^{-1}}$. Solid lines represent simulations with passive aerosols and dashed lines those with aerosol processing. The grey horizontal line indicates the location of the $0\,^\circ\mathrm{C}$ line in the different simulations.

[Figure]

**Figure 20.** Distribution of cloud top height in the different simulations for the time period between (a) 9 UTC to 12 UTC and (b) 12 UTC to 19 UTC, respectively. Cloud top is defined as the highest point in each column where the condensate content exceeds $1 \, \text{mg} \, \text{kg}^{-1}$.

[Figure]

**Figure 21.** Panels (a, b) show the change in accumulated precipitation relative to the simulations with standard aerosol profile for different $w_{max}$. Panels (c, d) show the change in condensed water path including cloud droplets, ice, and snow and (e, f) in the condensed water path including snow rain and graupel. All changes are relative to the simulations with the standard aerosol profiel. The panels in the left column are for the time period 9 - 12 UTC and those in the right for for 12 - 20 UTC. Results from simulations with higher (lower) aerosol loading are depicted in green (blue). Solid lines correspond to simulations with passive aerosol, dashed lines to simulations with aerosol processing.

---

## Referee Report (RR1)

I appreciate the efforts that the authors have gone to in order to respond to my comments. My comments have all been addressed very well and this version of the manuscript is much improved. In particular, the paper has been restructured to present the discussion of aerosol processing and aerosol-induced changes in the cloud field in separate sections. This greatly improves the flow of the argument and the clarity of the paper to the reader.

I recommend publication after addressing the few very minor comments below:

General comment: it would be worth including a very short (e.g. single sentence) description of the difference between bin vs bulk microphysics schemes (especially as you introduce CASIM as a new bulk scheme), before listing which of the cited studies use bin schemes and which use bulk schemes.

Revised manuscript:
P1 L2: heigh => height
P3 L11-13: "In addition..." – this sentence is incomplete?
P4 L4: "satellite data, that" – delete comma
P4 L15: sea => Sea
P4 L25 & L34: inconsistent spelling of focuses / focusses
P10 L4: "Another possibility are" => "Another possibility is"
P13 L21: waterpath => water path
P14 L28: waterpath => water path
P15 Lines 1,3,4,5,6,8,9,10: waterpath => water path
P15 L4, L5: rate => rates
P18 L18: "a results" => "a result"
P19 L25: "associated to" => "associated with"
P19 L25: "hypothesis" => "hypothesise"
P20 L 14: "may point to either with" => "may point either to issues with"
P20 L26: heigh => high
P22 L21: "smaller, if" => "smaller if"
Fig. 5: waterpath => water path
Fig. 7 L1, L3: waterpath => water path
Fig. A1: "For example illustrated with the green shaded area assumes a precipitation efficiency" => "For example the green shaded area illustrates an assumed precipitation efficiency"

Supplementary Information:
Fig. 3: groudn => ground
Fig. 9: profilesfrom => profiles from
Fig. 10: "on an air parcel" or "on air parcels"
Fig. 17: "certain rate" => "certain rates"
Fig. 19: I can't see a grey horizontal line on any of the profiles?

---

## Referee Report (RR2)

**Review of "Aerosol-cloud interactions in mixed-phase convective clouds. Part 1: Aerosol perturbations" – Miltenberger et al. (2018)**

In general, I am pleased with the responses provided by the authors and the corresponding changes made to the manuscript (exceptions provided below include the first Minor Concern and grammatical issues that remain causing the text to be hard to follow at times; there is also one instance where units are missing from a figure). Overall, I think the manuscript has been largely improved and should be accepted for publication pending the minor changes requested below.

**Minor Concerns**

1. In my original review, I questioned the use of aerosol number density as a prognostic variable because it is not a conserved quantity. My understanding is that if the variable is prognostic, then it is treated like all other scalars in the model and undergoes advection. Thus, it should be conserved. However, mixing ratios are commonly used instead of densities because they are conserved. Perhaps this is just a wording issue but should be clarified.

2. Grammatical issues remain in the text, especially regarding punctuation (in particular commas) that cause the text to be difficult to read. I found myself again having to read many sentences several times to fully understand the meaning. The main issue is the lack of commas in introductory phrases, e.g., "for the highest aerosol scenarios no further increase in the condensate production occurs, as clouds grow into an upper level stable layer". A comma needs to follow "scenarios." There are approximately 50 instances of this in the text, which is far too many to list in a review. I strongly encourage the authors to read the text carefully and perhaps have it reviewed by a non-author for guidance. Along the same lines, commas should be removed before phrases that begin with "if" and "when". Moreover, colons are used in many places in which they are not needed.

   Furthermore, hyphens are needed in many cases, particularly with compound adjectives, e.g., "aerosol-induced changes", "wing-mounted", or "high-resolution simulations".

   Issues with subject-verb agreement also remain. For example, "data" is plural; however, singular verbs are used with the term throughout the paper.

**Other Concerns**

1. The use of "knock-on effect" appears to be a British English phrase that may be largely unknown to the general scientific community. Consider using another phrase that would be understood by a general international audience.
2. When referring to prior works, please consider using the present or past tense consistently to avoid any confusion, e.g., "Smith (1990) found that…" or "Smith (1990) find that…"
3. Please define "COPE" the first time it is used in the abstract and main text.
4. In general, please be consistent with either "south-western" or "southwestern".
5. Please use "period" or a similar term for the two different periods discussed in the paper instead of "phase" to avoid confusion between the two phases of condensed water.
6. Page 3, Line 30: Remove "often".
7. Page 4, Line 10: It is not clear why "how large?" is in parentheses. Consider expaning on this or omitting.

8. Page 8, Line 21: Add "the" before "standard".
9. Page 9, Line 25: Add "the" before "domain".
10. Page 9, Line 35: Add "the" before "observed".
11. Page 10, Line 16: Change "which" to "that".
12. Page 11, Line 11: Add "the" before "simulations".
13. Page 11, Line 22: What about changes in chemistry and size?
14. Page 11, Line 30: Change "that" to "which".
15. Page 12, Lines 27-28: Please expand upon this statement because it seems as though the assumption is that aerosol particles are recycled back into clouds.
16. Page 13, Line 27: Reduction of what? The occurrence of cloud tops between 3 and 4 km?
17. Page 13, Line 31: Add "the" before "thermodynamic".
18. Page 14, Line 7: Change "in" to "into".
19. Page 14, Lines 17-19: This sentence needs to be reworded. Perhaps the confusion is because of the word "sensitivity" when in fact you mean a "response"?
20. Page 15, Line 6: Change "is evident also" to "is also evident".
21. Page 15, Line 16: Change "to" to "in".
22. Page 16, Line 17: Put "PE" in parenthesis and consider using it for the remainder of the text for conciseness.
23. Page 16, Line 31: Change "in" to "into".
24. Page 18, Line 8: Use "concentrations" for consistency.
25. Page 18, Line 18: Do you mean "decreasing aerosol number concentration"?
26. Page 19, Line 25: Change "to" to "with"?
27. Page 19, Line 28: Add "the" before "precipitation".
28. Page 20, Lines 12-13: Reword – perhaps just removing "profiles" and "structure" from the first two items in the list and then making the last word plural would suffice.
29. Page 20, Line 26: Please change "heigh" to "high".
30. Page 20, Line 31: Add "the" before "aerosol".
31. Page 21, Lines 6-7: Define the acronyms.
32. Page 21, Lines 31-32: This is an important point that I believe should be emphasized as well as placed in the context of prior studies that have suggested a similar finding (see Lebo (2017) and references therein).
33. Page 22, Line 21: The reference is not correct; please use Lebo et al. (2012).
34. Figure 4: Please add °W or similar to the x-axes of the figures.
35. Figure 6: Why did you choose to use $10^6$ kg for precipitation? Why not use domain average values in mm?

**References**

Lebo, Z. J. (2017), A numerical investigation on the effects of enhanced latent heat release in deep convective clouds relative to other factors, *J. Atmos. Sci.*, doi:10.1175/JAS-D-16-0368.1 (in press).

Lebo, Z. J., H. Morrison, and J. H. Seinfeld (2012), Are simulated aerosol-induced effects on deep convective clouds strongly dependent on saturation adjustment?, *Atmos. Chem. Phys.*, **12**, 9941-9964, doi:10.5194/acp-12-9941-2012.

---

## Author Response (AR2)

**Replies to review RC1**

General comment: it would be worth including a very short (e.g. single sentence) description of the difference between bin vs bulk microphysics schemes (especially as you introduce CASIM as a new bulk scheme), before listing which of the cited studies use bin schemes and which use bulk schemes.

*Change to paper:* We have added a sentence introducing bin and bulk microphysics scheme on p. 3, lines 17-21.

Revised manuscript:
P1 L2: heigh => height
P3 L11-13: "In addition..." – this sentence is incomplete?
P4 L4: "satellite data, that" – delete comma
P4 L15: sea => Sea
P4 L25 & L34: inconsistent spelling of focuses / focusses
P10 L4: "Another possibility are" => "Another possibility is"
P13 L21: waterpath => water path
P14 L28: waterpath => water path
P15 Lines 1,3,4,5,6,8,9,10: waterpath => water path
P15 L4, L5: rate => rates
P18 L18: "a results" => "a result"
P19 L25: "associated to" => "associated with"
P19 L25: "hypothesis" => "hypothesise"
P20 L 14: "may point to either with" => "may point either to issues with"
P20 L26: heigh => high
P22 L21: "smaller, if" => "smaller if"
Fig. 5: waterpath => water path
Fig. 7 L1, L3: waterpath => water path
Fig. A1: "For example illustrated with the green shaded area assumes a precipitation efficiency" => "For example the green shaded area illustrates an assumed precipitation efficiency"
Supplementary Information:
Fig. 3: groudn => ground
Fig. 9: profilesfrom => profiles from
Fig. 10: "on an air parcel" or "on air parcels"
Fig. 17: "certain rate" => "certain rates"
Fig. 19: I can't see a grey horizontal line on any of the profiles?

*Change to paper:* Thank you for pointing these issues out. They have been addressed as suggested.

**Replies to review RC2**

**Minor Concerns**

1. In my original review, I questioned the use of aerosol number density as a prognostic variable because it is not a conserved quantity. My understanding is that if the variable is prognostic, then it is treated like all other scalars in the model and undergoes advection. Thus, it should be conserved. However, mixing ratios are commonly used instead of densities because they are conserved. Perhaps this is just a wording issue but should be clarified.

*Change to paper: Thank you for posting this out. The prognostic variable in the model is of course number mass mixing ratio. We changed the text accordingly.*

2. Grammatical issues remain in the text, especially regarding punctuation (in particular commas) that cause the text to be difficult to read. I found myself again having to read many sentences several times to fully understand the meaning. The main issue is the lack of commas in introductory phrases, e.g., "for the highest aerosol scenarios no further increase in the condensate production occurs, as clouds grow into an upper level stable layer". A comma needs to follow "scenarios." There are approximately 50 instances of this in the text, which is far too many to list in a review. I strongly encourage the authors to read the text carefully and perhaps have it reviewed by a non-author for guidance. Along the same lines, commas should be removed before phrases that begin with "if" and "when". Moreover, colons are used in many places in which they are not needed.

Furthermore, hyphens are needed in many cases, particularly with compound adjectives, e.g., "aerosol-induced changes", "wing-mounted", or "high-resolution simulations".

Issues with subject-verb agreement also remain. For example, "data" is plural; however, singular verbs are used with the term throughout the paper.

*Change to paper: We have re-read the text again very carefully and corrected the raised issues.*

**Other Concerns**

1. The use of "knock-on effect" appears to be a British English phrase that may be largely unknown to the general scientific community. Consider using another phrase that would be understood by a general international audience.

*Change to paper: We reworded the sentence to avoid using this term: „This initial change in cloud droplet number should subsequently impact radiative and cloud microphysical processes …" (p. 2, l. 12/13).*

2. When referring to prior works, please consider using the present or past tense consistently to avoid any confusion, e.g., "Smith (1990) found that..." or "Smith (1990) find that..."

3. Please define "COPE" the first time it is used in the abstract and main text.

4. In general, please be consistent with either "south-western" or "southwestern".

5. Please use "period" or a similar term for the two different periods discussed in the paper instead of "phase" to avoid confusion between the two phases of condensed water.

6. Page 3, Line 30: Remove "often".

7. Page 4, Line 10: It is not clear why "how large?" is in parentheses. Consider expaning on this or omitting.

8. Page 8, Line 21: Add "the" before "standard".

9. Page 9, Line 25: Add "the" before "domain".

10. Page 9, Line 35: Add "the" before "observed".

11. Page 10, Line 16: Change "which" to "that".

12. Page 11, Line 11: Add "the" before "simulations".

*Change to paper: Thank you for pointing these issues out. They have been addressed as suggested.*

13. Page 11, Line 22: What about changes in chemistry and size?

*Change to paper: We added these to the list of potential changes and clarified that changes to chemistry are not included in CASIM (p. 11, lines 28-30).*

14. Page 11, Line 30: Change "that" to "which".
*Change to paper: Changed as suggested.*

15. Page 12, Lines 27-28: Please expand upon this statement because it seems as though the assumption is that aerosol particles are recycled back into clouds.
*Change to paper: We added some further explanation on p. 13, lines 1-3.*

16. Page 13, Line 27: Reduction of what? The occurrence of cloud tops between 3 and 4 km?
17. Page 13, Line 31: Add "the" before "thermodynamic".
18. Page 14, Line 7: Change "in" to "into".
19. Page 14, Lines 17-19: This sentence needs to be reworded. Perhaps the confusion is because of the word "sensitivity" when in fact you mean a "response"?
20. Page 15, Line 6: Change "is evident also" to "is also evident".
21. Page 15, Line 16: Change "to" to "in".
22. Page 16, Line 17: Put "PE" in parenthesis and consider using it for the remainder of the text for conciseness.
23. Page 16, Line 31: Change "in" to "into".
24. Page 18, Line 8: Use "concentrations" for consistency.
25. Page 18, Line 18: Do you mean "decreasing aerosol number concentration"?
26. Page 19, Line 25: Change "to" to "with"?
27. Page 19, Line 28: Add "the" before "precipitation".
28. Page 20, Lines 12-13: Reword – perhaps just removing "profiles" and "structure" from the first two items in the list and then making the last word plural would suffice.
29. Page 20, Line 26: Please change "heigh" to "high".
30. Page 20, Line 31: Add "the" before "aerosol".
31. Page 21, Lines 6-7: Define the acronyms.
*Change to paper: Thank you for pointing these issues out. They have been addressed as suggested.*

32. Page 21, Lines 31-32: This is an important point that I believe should be emphasized as well as placed in the context of prior studies that have suggested a similar finding (see Lebo (2017) and references therein).
*Change to paper: Thank you for pointing this paper. We added some discussion of the suggested references on p. 22, lines 11-18.*

33. Page 22, Line 21: The reference is not correct; please use Lebo et al. (2012).
34. Figure 4: Please add °W or similar to the x-axes of the figures.
*Change to paper: Thank you for pointing these issues out. They have been addressed as suggested.*

35. Figure 6: Why did you choose to use $10^6$ kg for precipitation? Why not use domain average values in mm?
*Reply: We use this unit for better comparison to the other plots pertaining to the condensate mass budget, which is the essential part of the paper.*